# pFL-Bench: A Comprehensive Benchmark for Personalized Federated Learning

**Daoyuan Chen, Dawei Gao, Weirui Kuang, Yaliang Li**[*] **Bolin Ding**
Alibaba Group
{daoyuanchen.cdy, gaodawei.gdw, weirui.kwr}@alibaba-inc.com
{yaliang.li, bolin.ding}@alibaba-inc.com

## Abstract

Personalized Federated Learning (pFL), which utilizes and deploys distinct local models, has gained increasing attention in recent years due to its success in handling the statistical heterogeneity of FL clients. However, standardized evaluation and systematical analysis of diverse pFL methods remain a challenge. Firstly, the highly varied datasets, FL simulation settings and pFL implementations prevent easy and fair comparisons of pFL methods. Secondly, the current pFL literature diverges in the adopted evaluation and ablation protocols. Finally, the effectiveness and robustness of pFL methods are under-explored in various practical scenarios, such as the generalization to new clients and the participation of resource-limited clients. To tackle these challenges, we propose the first comprehensive pFL benchmark, pFL-Bench, for facilitating rapid, reproducible, standardized and thorough pFL evaluation. The proposed benchmark contains more than 10 dataset variants in various application domains with a unified data partition and realistic heterogeneous settings; a modularized and easy-to-extend pFL codebase with more than 20 competitive pFL method implementations; and systematic evaluations under containerized environments in terms of generalization, fairness, system overhead, and convergence. We highlight the benefits and potential of state-of-the-art pFL methods and hope the pFL-Bench enables further pFL research and broad applications that would otherwise be difficult owing to the absence of a dedicated benchmark. The code is released at https://github.com/alibaba/FederatedScope/tree/master/benchmark/pFL-Bench. [1]

## 1 Introduction

Federated learning (FL) is an emerging machine learning (ML) paradigm, which collaboratively trains models via coordinating certain distributed clients (*e.g.*, smart IoT devices) with a logically centralized aggregator [1, 2]. Due to the benefit that it does not transmit local data and circumvents the high cost and privacy risks of collecting raw sensitive data from clients, FL has gained widespread interest and has been applied in numerous ML tasks such as image classification [3, 4], object detection [5, 6], keyboard suggestion [7, 8], text classification [9], relation extraction [10], speech recognition [11, 12], graph classification [13], recommendation [14, 15], and healthcare [16, 17].

Pioneering FL researchers made great efforts to find a global model that performs well for most FL clients [18, 19, 20, 21]. However, the intrinsic statistical and system heterogeneity of clients limits the performance and applicability of such classical FL methods [22, 23]. Taking the concept shift case as an example, in which the conditional distribution $P(Y|X)$ varies across some clients who have the same marginal distributions $P(X)$ [18], a shared global model cannot fit these clients well

---

[*]corresponding author

[1]We will continuously maintain the benchmark and update the codebase and arXiv version.

36th Conference on Neural Information Processing Systems (NeurIPS 2022) Track on Datasets and Benchmarks.

at the same time. Recently, the personalized FL (pFL) methods that utilize client-distinct models to overcome these challenges have been gaining increasing popularity, such as those based on multi-task learning [24, 25, 26], meta-learning [27, 28], and transfer learning [29, 30, 31].

Even though fruitful pFL methods have been explored, a standard benchmark is still lacking. As a result, the evaluation of pFL methods is currently with non-standardized datasets and implementations, highly diverse evaluation protocols, and unclear effectiveness and robustness of pFL methods under various practical scenarios. To be specific:

- **Non-standardized datasets and implementations for pFL.** Currently, researchers often use custom FL datasets and implementations to evaluate the effectiveness of proposed methods due to the absence of standardized pFL benchmarks. For example, although many pFL works use the same public FEMNIST [32] and CIFAR-10/100 [33] datasets, the partition manners can be divergent: the number of clients is 205 in [26] while 539 in [34] for FEMNIST; and [35] adopts the Dirichlet distribution based partition while [36] uses the pathological partition for CIFAR-10/100. Prior pFL studies set up different computation environments and simulation settings, increasing the difficulty of fast evaluation and the risk of unfair comparisons.

- **Diverse evaluation protocols.** The current pFL methods often focus on different views and adopt diverse evaluation protocols, which may lead to isolated development of pFL and prevent pFL research from reaching its full potential. For example, besides the global accuracy improvement, a few works studied the local accuracy evaluation characterized by fairness, and system efficiency in terms of communication and computational costs [26, 37]. Without a careful design and control of the evaluation, it is difficult to compare the pros and cons of different pFL methods and understand how much costs we pay for the personalization.

- **Under-explored practicability of pFL in various scenarios.** Most existing pFL methods examine their effectiveness in several mild Non-IID FL cases [23]. However, it is unclear whether existing pFL methods can consistently work well in more practical scenarios, such as the participation of partial clients in which the clients have spotty connectivity [38]; the participation of resource-limited clients in which the personalization is required to be highly efficient [39]; and the generalization to new clients in which learned models will be applied to new clients that do not participate in the FL process [40].

To quantify the progress in the pFL community and facilitate rapid, reproducible, and generalizable pFL research, we propose the first comprehensive pFL benchmark characterized as follows:

- We provide 4 benchmark families with 12 dataset variants for diverse application domains involving image, text, graph and recommendation data, each with unified data partition and some realistic Non-IID settings such as clients sampling and the participation of new clients. Some public popular DataZoos such as LEAF [32], Torchvision [41] and Huggingface datasets [42] are also compatible to the proposed pFL-Bench to enable flexible and easily-extended experiments.

- We implement an extensible open-sourced pFL codebase that contains more than 20 pFL methods, providing fruitful state-of-the-art (SOTA) methods re-implementations, unified interfaces and pluggable personalization sub-modules such as model parameter decoupling, model mixture, meta-learning and personalized regularization.

- We conduct systematic evaluation under unified experimental settings and containerized environments to show the efficacy of pFL-Bench and provide standardized scripts to ensure the reproducibility and maintainability of pFL-Bench. We also highlight the advantages of pFL methods and opportunities for further pFL study in terms of generalization, fairness, system overhead and convergence.

## 2  Related Works

**Personalized Federated Learning.**  Despite the promising performance using a shared global model for all clients as demonstrated in [1, 43, 44, 45, 46, 47], it is challenging to find a converged best-for-all global model under statistical heterogeneity among clients [48, 49, 50]. As a natural way to handle the heterogeneity, personalization is gaining popularity in recent years. Fruitful pFL literatures have explored the accuracy and convergence improvement based on clustering [51, 48, 52], multi-task learning [53, 54, 55, 56], model mixture [36, 26, 57, 58], model parameter

decoupling [37, 6], Bayesian treatment [59, 54], knowledge distillation [60, 31, 61], meta-learning [62, 63, 62, 28, 64, 65], and transfer learning [66, 67, 68]. We refer readers to related FL and pFL survey papers for more details [18, 22, 23]. In pFL-Bench, we provide modularized re-implementation for numerous SOTA pFL methods with several fundamental and pluggable sub-routines for easy and fast pFL research and deployment. We plan to add more pFL methods in the future and also welcome contributions to the pFL-Bench.

**Federated Learning Benchmark.**   We are aware that there are great efforts on benchmarking FL from various aspects, such as heterogeneous datasets (LEAF [32], TFF [69]), heterogeneous system resources (Flower [70], FedML [71], FedScale [38]), and specific domains (FedNLP [72], FS-G [73]). However, they mostly benchmarked general FL algorithms, lacking recently proposed pFL methods that perform well on heterogeneous FL scenarios. Besides, no benchmark so far supports the evaluation of generalization to new clients; and few existing benchmarks simultaneously support comprehensive evaluation for trade-offs among accuracy, fairness and system costs. We hope to close these gaps with this proposed pFL-Bench, and facilitate further pFL research and broad applications.

## 3   Background and Problem Formulation

### 3.1   A Generalized FL Setting

We first introduce some important concepts in FL, taking the FedAvg method [1] as an illustrative example. A typical FL procedure using FedAvg is as follows: Each client $i \in \mathcal{C}$ has its own private dataset $\mathcal{D}_i$ over $\mathcal{X} \times \mathcal{Y}$, and the goal of FL is to train a single global model $\theta_g$ with collaborative learning from this set of clients $\mathcal{C}$ without directly sharing their local data. At each FL round, the server broadcasts $\theta_g$ to selected clients $\mathcal{C}_s \subseteq \mathcal{C}$, who then perform local learning based on the private local data and upload the local update information (*e.g.*, the gradients of trained models) to the server. After collecting and averagely aggregating the update information from clients, the server applies the updates into $\theta_g$ for the next-round federation and the process repeats.

Then we present a generalized FL formulation, which establishes the proposed comprehensive benchmark in terms of diverse evaluation and personalization perspectives. Specifically, besides the FL-participated clients $\mathcal{C}$, we consider a set of new clients that do not participate in the FL training process and denote it as $\tilde{\mathcal{C}}$. Most FL approaches implicitly solve the following problem:

$$\min_{\{h_{\theta_g}\} \cup \{h_{\theta_i}\}_{i \in \mathcal{C}} \cup \{h_{\theta_j}\}_{j \in \tilde{\mathcal{C}}}} \alpha G\big(\{\mathbb{E}_{(x,y) \sim \mathcal{D}_i}[f(\theta_g; x, y)]\}_{i \in \mathcal{C}}\big) + \beta L\big(\{\mathbb{E}_{(x,y) \sim \mathcal{D}_i}[f(\theta_i; x, y)]\}_{i \in \mathcal{C}}\big)$$
$$+ \gamma R\big(\{\mathbb{E}_{(x,y) \sim \mathcal{D}_j}[f(\theta_j; x, y)|\theta_g]\}_{j \in \tilde{\mathcal{C}}}\big) + \zeta Q(\theta_g, \theta_k)_{k \in (\mathcal{C} \cup \tilde{\mathcal{C}})},$$
(1)

where $f(\theta; x, y)$ indicates the loss at data point $(x, y)$ with model $\theta$. The term $G(\cdot)$ indicates the global objective based on the shared global model $\theta_g = Agg([\theta_i]_{i \in \mathcal{C}})$ with an aggregation function $Agg(\cdot)$ for model parameters, and $L(\cdot)$ indicates the local objective based on the local distinct models $[\theta_i]_{i \in \mathcal{C}}$. We note that $G(\cdot)$ and $L(\cdot)$ can be in various forms such as uniform averaging or weighted averaging according to local training data size of clients, which corresponds to the commonly used intra-client generalization case [40].

For the latter two terms, $R(\cdot)$ usually has similar forms to $G(\cdot)$ and measures the generalization to new clients $\tilde{\mathcal{C}}$ that do not contribute to the FL training process. $Q(\cdot)$ indicates the modeling of the relationship between the global and local models, such as the $L^2$ norm to regularize the model parameters in Ditto [26]. Besides, different pFL methods may flexibly introduce various constraints on this optimization objective, which we have omitted in Eq.(1) for brevity. The coefficients $\alpha, \beta, \gamma$ and $\zeta$ trade off these terms. For non-personalized FL algorithms, $\beta = 0$. Although the generalization term $R(\cdot)$ has been primarily explored in a few recent studies [40, 35], most existing FL works overlooked it with $\gamma = 0$. Later, we will discuss more instantiations for $L(\cdot)$ and $Q(\cdot)$ in the personalization setting.

### 3.2   Personalization Setup

With the above generalized formulation, we can see that existing pFL works achieve personalization via multi-granularity objects, including the global model $\theta_g$ and local models $[\theta_i]$. For example,

Table 1: Statistics of the experimental datasets, tasks, and models in pFL-bench. We sample 5% clients from FEMNIST [32]. Following previous works [71, 56, 37, 28], we adopt Dirichlet allocation with different $\alpha$s to simulate the heterogeneous partition for CIFAR10 and textual datasets. The $\mu$ and $\sigma$ indicate the mean and std of the number of samples per client. More datasets from popular Datazoos such as LEAF [32], Torchvision [41], Huggingface datasets [42] and FederatedScope-GNN [73] are also supported. Detailed descriptions can be found in Appendix A.

| Dataset | Task | Model | Partition By | # Clients | # Sample Per Client | |
|---|---|---|---|---|---|---|
| FEMNIST | | | Writers | 200 | $\mu$=217 | $\sigma$=73 |
| CIFAR10-$\alpha$5 | | | | 100 | $\mu$=600 | $\sigma$=46 |
| CIFAR10-$\alpha$0.5 | Image Classification | CNN | Labels | 100 | $\mu$=600 | $\sigma$=137 |
| CIFAR10-$\alpha$0.1 | | | | 100 | $\mu$=600 | $\sigma$=383 |
| COLA | Linguistic Acceptability | BERT | Labels | 50 | $\mu$=192 | $\sigma$=159 |
| SST-2 | Sentiment Analysis | | | 50 | $\mu$=1,364 | $\sigma$=1,291 |
| Twitter | Sentiment Analysis | LR | Users | 13,203 | $\mu$=10 | $\sigma$=11 |
| Cora | | | | 5 | $\mu$=542 | $\sigma$=30 |
| Pubmed | Node Classification | GIN | Community | 5 | $\mu$=3,943 | $\sigma$=34 |
| Citeseer | | | | 5 | $\mu$=665 | $\sigma$=29 |
| MovieLens1M | Recommendation | MF | Users | 1000 | $\mu$=1,000 | $\sigma$=482 |
| MovieLens10M | | | Items | 1000 | $\mu$=10,000 | $\sigma$=8,155 |

many two-step pFL approaches first find a strong $\theta_g$ in the FL training stages, then get local models $[\theta_i]$ by fine-tuning $\theta_g$ on local data and use $[\theta_i]$ in inference [23, 74]. A more flexible manner is to directly learn distinct local models $[\theta_i]$ in the FL process, while this introduces additional storage and computation costs for clients [75]. Recently, to gain a better accuracy-efficiency trade-off, several pFL works propose to only personalize a sub-module $\pi_i$ of $\theta_i$, and transmit and aggregate the remaining part as $\theta_g = Agg([\theta_i \setminus \pi_i]_{i \in \mathcal{C}})$ [24].

We illustrate some representative personalization operations in pFL works w.r.t. the different choices of the local objective $L(\cdot)$ and $Q(\cdot)$. Fine-tuning is a basic step widely used in abundant pFL works [63] to minimize $\mathbb{E}_{(x,y) \sim \mathcal{D}_i}[f(\theta_i; x, y)]$, where $\theta_i$ is usually initialized using the parameters of $\theta_g$ before fine-tuning. Model mixture is a general pFL approach assuming that the local data distribution is a mixture of $K$ underlying data distributions $\mathcal{D}_i = Mix(\mathcal{D}_{i,k}, w_{i,k})$ with mixture weight $w_{i,k}$ for $k \in [K]$ [56], thus learning a group of intermediate models is suitable to handle the data heterogeneity as $\theta_i = Mix(\theta_{i,k})_{k \in [K]}$. For clustering-based pFL methods, $K$ indicates the cluster number, and the mixture weight is 0-1 indicative function for belonged clusters [25]. Besides, taking $\theta_g$ as the reference point, model interpolation $\theta_i \equiv w_g\theta_g + (1 - w_g)\theta_i$ and model regularization $\theta_i = argmin\left(\sum_{(x,y) \sim \mathcal{D}_i}(f(\theta_i; x, y) + \frac{\lambda}{2}||\theta_i - \theta_g||)\right)$ are also widely explored in the pFL literature [76], where $\lambda$ is the regularization factor.

## 4 Benchmark Design and Resources

### 4.1 Datasets and Models

We conduct experiments on 12 publicly available dataset variants with heterogeneous partition in our benchmark. These datasets are popular in the corresponding fields, and cover a wide range of domains, scales, partition manners and Non-IID degrees. We list the statistics in Table 1 and illustrate the violin plot of data size per client in Figure 1, which show diverse properties across the FL datasets, enabling thorough comparisons among different pFL methods. We provide more detailed descriptions of these datasets in Appendix A. Besides, with a carefully designed modularity, our code-base is compatible with a large number of datasets from other public popular DataZoos, including LEAF [32], Torchvision [41], Huggingface datasets [42] and FederatedScope-GNN [73].

We preset the widely adopted CNN model [77, 78, 79, 80] with additional batch normalization layers for image datasets, and the pre-trained BERT-Tiny model from [81] and linear regression (LR) model for the textual datasets. For the graph and recommendation datasets, we preset the graph isomorphism neural network, GIN [82], and Matrix Factorization (MF) [83] respectively. It is worth noting that pFL-Bench provides a unified model interface decoupled with FL algorithms, enabling users to easily

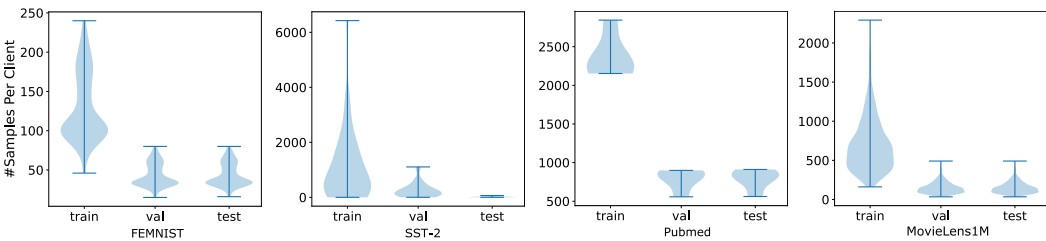

Figure 1: The violin plot of the number of samples per client for partially adopted datasets. In Appendix A, we present the plots for other datasets, the label skew visualization and clients' pairwise similarity of label distribution in terms of Jensen–Shannon distance.

register and use more customized models or built-in models from existing ModelZoos including Torchvision [41], Huggingface [84], and FederatedScope [13].

**Benchmark scenarios.**   (GENERALIZATION) For a comprehensive evaluation, pFL-Bench supports examining the generalization performance for both FL-participated and FL-non-participated clients, *i.e.*, the $R(\cdot)$ term in formulation (1). Specifically, we randomly select 20% clients as non-participated clients for each dataset, and these clients will not transmit their training-related messages during the FL processes.

(CLIENT SAMPLING) In addition to generalization performance, we also care about how the pFL methods perform when adopting client sampling in FL processes, which is useful in cross-device scenarios where a large number of clients have spotty connectivity. For the image, text and recommendation datasets, we uniformly sample 20% clients without replacement from the participating clients at each FL round.

(CROSS-SILO V.S. CROSS-DEVICE) Note that the adopted datasets have quite different numbers of clients after heterogeneous partition. We choose the three graph datasets with small numbers of clients to simulate cross-silo FL scenarios [73, 55], while the other datasets correspond to different scales of cross-device FL scenarios.

## 4.2   Methods

We consider abundant methods for extensive pFL comparisons. The pFL-bench provides unified and modularized interfaces for a range of popular and SOTA methods in the following three categories:
**Non-pFL methods.** As two naive methods sitting on opposite ends of the local-global spectrum, we evaluate the *Global-Train* method that trains the model from centralized data merged from all clients, and *Isolated* method that trains a separated model for each client without any information transmission among clients. Besides, we consider the classical *FedAvg* [1] with weighted averaging based on local data size, the *FedProx* [85] that introduces proximal term during the local training process, and *FedOpt* [43] that generalizes FedAvg by introducing an optimizer for the FL server.

**pFL methods.** We compare several SOTA methods including *FedBN* [86] that is a simple yet effective method to deal with feature shift Non-IID, via locally maintaining the clients' batch normalization parameters without transmitting and aggregation; *Ditto* [26] that improves fairness and robustness of FL by training local personalized model and global model simultaneously, in which the local model update is based on regularization to global model parameters; *pFedMe* [78] that is a meta-learning based method and decouples personalized model and global model with Moreau envelops; *HypCluster* [74] that splits users into clusters and learns different models for different clusters; *FedEM* [56] that assumes local data distribution is a mixture of unknown underlying distributions, and correspondingly learns a mixture of multiple intermediate models with Expectation-Maximization algorithm.

**Combined variants**. Note that pFL-Bench provides pluggable re-implementations of existing methods, enabling users can pick different personalized behaviors and different personalized objects to form a new pFL variant. Here we combine *FedBN*, *FedOpt*, and *Fine-tuning (FT)* with other compatible methods, and provide fine-grained ablations via more than 20 method variants for systematic pFL study and explorations of pFL potential. More details of the considered methods are in Appendix B.

### 4.3 Evaluation Criteria

We propose a unified and comprehensive evaluation protocol in pFL-Bench, in which evaluations from multiple perspectives are taken into consideration. (1) For the **generalization** examination, we support monitoring on both the server and client sides, with various and extensible metric aggregation manners, and a wide range of metrics for diverse tasks such as classification, regression and ranking. (2) We also report several **fairness-related** metrics, including standard deviation, and the top and bottom deciles of performance across different clients. (3) Numerous **systematical metrics** are considered as well, including the process running time, the memory cost w.r.t. average and peak memory usage, the total computational cost w.r.t. FLOPs in server and clients, the communication cost w.r.t. the total number of downloaded/uploaded bytes, and the number of FL rounds to convergence. Detailed descriptions can be found in Appendix B.

### 4.4 Codebase

To facilitate the innovation for pFL community, our pFL-Bench contains a well-modularized, easy-to-extend codebase for standardization of implementation, evaluation, and ablation of pFL methods.

**pFL implementations.** We build the pFL-bench upon an event-driven FL framework FEDERATED-SCOPE (FS) [87], which abstracts FL information transmitting and processing as message passing and several pluggable subroutines. We eliminate the cumbersome engineering for coordinating FL participants with the help of FS, and customize many message handlers and subroutines, such as model parameter decoupling, model mixture, local fine-tuning, meta-learning and regularization for personalization. By combining these useful and pluggable components, we re-implement a number of SOTA pFL methods with unified and extensible interfaces. This modularity also makes the usage of pFL methods convenient, and makes the contribution of new pFL methods easy and flexible. We release the codes with Apache License 2.0 and will continuously include more pFL methods.

**Reproducibility.** To enable easily reproducible research, we conduct experiments in containerized environments and provide standardized and documented evaluation procedures for prescribed metrics. For fair comparisons, we search the optimal hyper-parameters using the validation sets for all methods, with early stopping and large numbers of total FL rounds. We run experiments 3 times with the optimal configurations and report the average results. All the experiments are conducted on a cluster of 8 Tesla V100 and 64 NVIDIA GTX 1080 Ti GPUs, taking $\sim$13,000 runs with a total of $\sim$112 days process computing time. More details, such as hyper-parameter search spaces, can be found in Appendix C.

## 5 Experimental Results and Analysis

To demonstrate the utility of pFL-Bench in providing fair, comprehensive, and rigorous comparisons among pFL methods, we conduct extensive experiments and present some main results in terms of generalization, fairness and efficiency under various FL datasets and scenarios. The complete experimental results are presented in Appendix D due to the space limitation.

### 5.1 Generalization

We present the accuracy results for both participated and un-participated clients in Table 2 on FEMNIST, SST-2 and PUBMED datasets, where $\overline{Acc}$ and $\widetilde{Acc}$ indicates the aggregated accuracy weighted by the local data samples of participated and un-participated clients respectively, and $\Delta = \widetilde{Acc} - \overline{Acc}$ indicates the generalization gap.

**Comparison of original methods.** For the original methods without combination ("-"), we mark the best and second-best results as red and blue respectively in Table 2. Notably, we find that *no method can consistently beat others across all metrics and all datasets*. The pFL methods gain significantly better $\overline{Acc}$ over FedAvg in some cases (*e.g.*, Ditto on FEMNIST), showing the effectiveness of personalization. However, the advantages of pFL methods on un-participated clients' generalization $\widetilde{Acc}$ are relatively smaller than those on intra-client generalization $\overline{Acc}$. The methods Ditto and

Table 2: Accuracy results for both participated clients and un-participated clients. $\overline{Acc}$ indicates the aggregated accuracy weighted by the number of local data samples of participated clients, $\widetilde{Acc}$ indicates the aggregated accuracy of un-participated clients, and $\Delta$ indicates the participation generalization gap. **Bold** and underlined indicate the best and second-best results among all compared methods, while red and blue indicate the best and second-best results for original methods without combination "-".

| | FEMNIST, $s = 0.2$ | | | SST-2 | | | PUBMED | | |
|---|---|---|---|---|---|---|---|---|---|
| | $\overline{Acc}$ | $\widetilde{Acc}$ | $\Delta$ | $\overline{Acc}$ | $\widetilde{Acc}$ | $\Delta$ | $\overline{Acc}$ | $\widetilde{Acc}$ | $\Delta$ |
| Global-Train | 74.51 | - | - | 80.57 | - | - | 87.01 | - | - |
| Isolated | 68.74 | - | - | 60.82 | - | - | 85.56 | - | - |
| FedAvg | 83.97 | 81.97 | -2.00 | 74.88 | 80.24 | 5.36 | 87.27 | 72.63 | -14.64 |
| FedAvg-FT | 86.44 | 84.94 | -1.50 | 74.14 | 83.28 | 9.13 | 87.21 | 79.78 | -7.43 |
| FedProx | 84.10 | 81.49 | -2.61 | 74.36 | 79.20 | 4.84 | 87.23 | 75.02 | -12.21 |
| FedProx-FT | 87.34 | 85.27 | -2.08 | 79.94 | 80.48 | 0.59 | 88.24 | 79.12 | -9.12 |
| pFedMe | 87.50 | 82.76 | -4.73 | 71.27 | 69.34 | -1.92 | 86.91 | 71.64 | -15.27 |
| pFedMe-FT | 88.19 | 82.46 | -5.73 | 75.61 | 66.48 | -9.13 | 85.71 | 77.07 | -8.64 |
| HypCluster | 83.80 | 81.88 | -1.92 | 46.26 | 61.32 | 15.05 | 87.20 | 75.37 | -11.83 |
| HypCluster-FT | 87.79 | 85.67 | -2.12 | 52.46 | 78.67 | 26.21 | 86.43 | 76.69 | -9.74 |
| FedBN | 86.72 | 7.86 | -78.86 | 74.88 | 75.40 | 0.52 | 88.49 | 52.53 | -35.95 |
| FedBN-FT | 88.51 | 82.87 | -5.64 | 68.81 | 82.43 | 13.63 | 87.45 | 80.36 | -7.09 |
| FedBN-FedOPT | 88.25 | 8.77 | -79.49 | 64.70 | 65.50 | 0.81 | 87.87 | 42.72 | -45.15 |
| FedBN-FedOPT-FT | 88.14 | 80.25 | -7.88 | 68.65 | 70.56 | 1.91 | 87.54 | 77.07 | -10.47 |
| Ditto | 88.39 | 2.20 | -86.19 | 52.03 | 46.79 | -5.24 | 87.27 | 2.84 | -84.43 |
| Ditto-FT | 85.72 | 56.96 | -28.76 | 56.49 | 65.50 | 9.01 | 87.47 | 35.03 | -52.44 |
| Ditto-FedBN | 88.94 | 2.20 | -86.74 | 56.03 | 46.79 | -9.24 | 88.18 | 2.84 | -85.34 |
| Ditto-FedBN-FT | 86.53 | 58.96 | -27.57 | 53.15 | 66.49 | 13.34 | 87.83 | 28.52 | -59.30 |
| Ditto-FedBN-FedOpt | 88.73 | 2.20 | -86.54 | 57.67 | 46.79 | -10.88 | 87.81 | 2.84 | -84.97 |
| Ditto-FedBN-FedOpt-FT | 87.02 | 55.22 | -31.80 | 52.89 | 66.49 | 13.60 | 87.60 | 18.18 | -69.42 |
| FedEM | 84.35 | 82.81 | -1.54 | 75.78 | 67.67 | -8.11 | 85.64 | 71.12 | -14.52 |
| FedEM-FT | 86.17 | 85.01 | -1.16 | 64.86 | 81.63 | 16.77 | 85.88 | 78.08 | -7.80 |
| FedEM-FedBN | 84.37 | 12.88 | -71.49 | 75.43 | 62.81 | -12.62 | 88.12 | 48.64 | -39.48 |
| FedEM-FedBN-FT | 88.29 | 83.96 | -4.33 | 64.96 | 81.04 | 16.08 | 86.38 | 72.02 | -14.35 |
| FedEM-FedBN-FedOPT | 82.12 | 6.64 | -75.48 | 72.25 | 64.69 | -7.56 | 87.56 | 42.37 | -45.19 |
| FedEM-FedBN-FedOPT-FT | 87.54 | 85.76 | -1.79 | 62.26 | 73.87 | 11.61 | 87.49 | 72.39 | -15.09 |

FedBN even fail to gain reasonable $\widetilde{Acc}$ on FEMNIST and PUBMED datasets, as these methods did not discuss how to apply to unseen clients, we have kept the behaviors of their algorithms in inference in order to respect the original method. And we examine their running dynamics and find that their local models diverge with the un-participated clients' data. Besides, there are still performance gaps between pFL methods and the Global-Train method on the textual dataset. These observations demonstrate plenty of room for pFL improvement.

**Comparison of pFL variants.** We then extend the comparison to include FL variant methods incorporating other compatible personalized operators or methods. In a nutshell, almost all the best and second-best results (marked as **Bold** and underlined) are achieved by the combined variants, showing the efficacy and flexibility of our benchmark, and the potential of new pFL methods. Among these methods, fine-tuning (FT) effectively improves both $\overline{Acc}$ and $\widetilde{Acc}$ metrics in most cases, even for pFL methods that have been implicitly adopted local training process, which calls for a deeper understanding about how much should we fit the local data and how the personalized fitting impacts the FL dynamics. Besides, FedBN is a simple yet effective method to improve $\overline{Acc}$ while may bring a negative effect on $\widetilde{Acc}$ (e.g., FedEM v.s. FedEM-FT), since the feature space of un-participated clients lacks informative characterization and the frozen BN parameters of the global model can arbitrarily mismatch these un-participated clients. We also find that FedBN is much more effective on graph data with the GIN model than those on textual data with the BERT model, in which we made a simple modification that filters out the Layer Normalization parameters, showing the opportunity of designing domain- and model-specific pFL techniques.

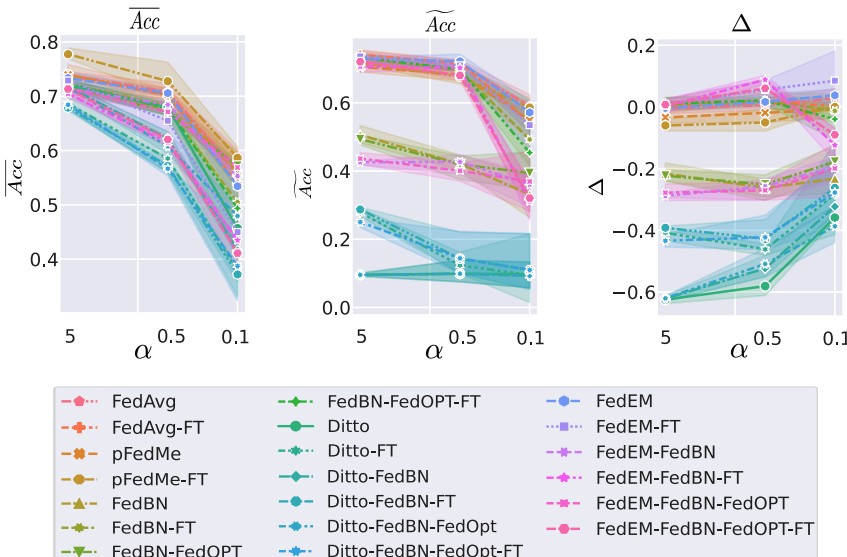

Figure 3: The accuracy results of participated, un-participated clients and generalization gap when varying Dirichlet factor $\alpha$ for CIFAR10 dataset. Results in table format are in Appendix D.1.

**Effect of Non-IID split.** To gain further insight into the pFL methods, we vary the Non-IID degree with different Dirichlet factor $\alpha$ for CIFAR10 dataset and illustrate the results in Figure 3. Generally speaking, almost all methods gain performance degradation as the Non-IID degree increases (from $\alpha$=5 to $\alpha$=0.1). Besides, we can see that most of pFL methods show superior accuracy and robustness over FedAvg especially for the highly heterogeneous case with $\alpha$=0.1. These results indicate the benefit of pFL methods in Non-IID situations, as well as their substantial space for improvement.

**Effect of client sampling.** We also vary the client sampling rate $s$ and present the generalization results on FEMNIST dataset in Figure 4. In summary, most pFL methods still achieve better results than FedAvg as $s$ decreases. However, the advantages are diminishing for un-participating clients and several pFL methods prune to fail with small $s$. In addition, we observe that FT increases the performance variance in client sampling cases. Although several pFL works provide convergence guarantees under mild assumptions, there remain open questions about the theoretical impact of clients with spotty connections [88] in the personalization case, and the design of robust pFL algorithms.

## 5.2 Fairness Study

We then empirically investigate what degrees of fairness can be achieved by pFL methods, and report the equally-weighted average of accuracy $\overline{Acc}'$, the standard deviation $\sigma$ across evaluated clients, and the bottom individual accuracy $\widecheck{Acc}$ in Table 3. These metrics are considered as the fairness criteria in related pFL works [26, 56]. We find that $\overline{Acc}'$ is usually smaller than the one weighted by local data size ($\overline{Acc}$ in Table 2), indicating the client bias in existing pFL evaluation. Across the three datasets, the $\sigma$s on SST-2 are much larger (at dozen-scales) than those on the image dataset ($6.33 \sim 11.02$), which are larger than those on the graph dataset ($3.06 \sim 6.26$), leaving room for further research to understand this difference and improve the fairness in various application domains. Interestingly, compared with FedAvg, pFL methods can effectively improve bottom accuracy, while they may gain larger standard deviations. An exception is Ditto-based methods on SST-2, as the parameter regularization in Ditto may fail for the complex BERT model. Besides, the Isolated method performs badly for clients having a very small data size (a few dozens), even gains $\widecheck{Acc}$=0 on SST-2. We find that most of the other methods achieve much better results than it, verifying the benefits of transmitting knowledge across clients.

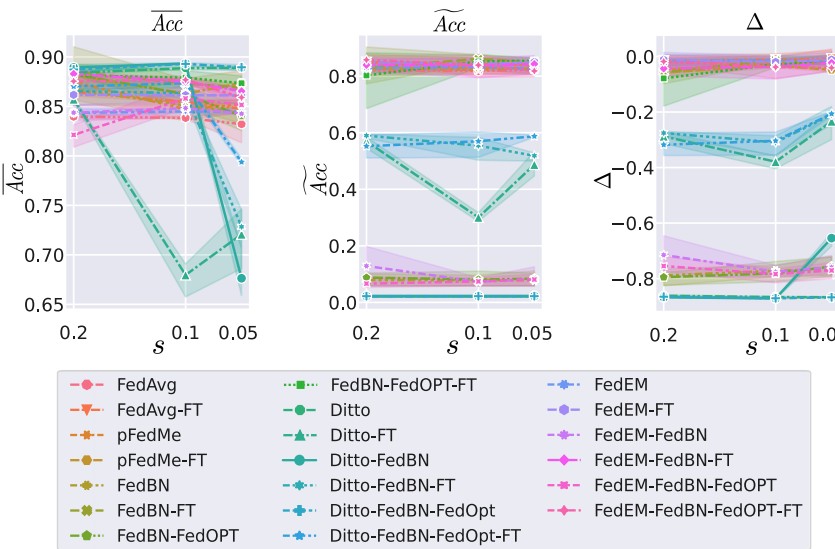

Figure 4: The accuracy results of participated, un-participated clients and generalization gap when varying the client sampling rate $s$ on FEMNIST dataset. Performance of some pFL methods degrades as $s$ decreases such as Ditto and Ditto-FT, calling for robustness improvements of pFL. Detailed results in table format can be found in D.1.

Table 3: Fairness results in terms of $\overline{Acc}'$ indicating the equally-weighted average, $\sigma$ indicating the standard deviation of the average accuracy, and $\widecheck{Acc}$ indicating the bottom accuracy. **Bold**, underlined, red and blue indicate the same highlights as used in Table 2.

| | FEMNIST, $s = 0.2$ | | | SST-2 | | | PUBMED | | |
|---|---|---|---|---|---|---|---|---|---|
| | $\overline{Acc}'$ | $\sigma$ | $\widecheck{Acc}$ | $\overline{Acc}'$ | $\sigma$ | $\widecheck{Acc}$ | $\overline{Acc}'$ | $\sigma$ | $\widecheck{Acc}$ |
| Isolated | 67.08 | 10.76 | 53.16 | 59.40 | 41.29 | 0.00 | 84.67 | 6.26 | 74.63 |
| FedAvg | 82.40 | 9.91 | 69.11 | 76.30 | 22.02 | 44.85 | 86.72 | 3.93 | 79.76 |
| FedAvg-FT | 85.17 | 8.69 | 72.34 | 75.36 | 27.67 | 31.08 | 86.71 | 3.86 | 80.57 |
| FedProx | 82.37 | 10.25 | 68.57 | 72.25 | 22.03 | 43.51 | **88.06** | 3.62 | 80.08 |
| FedProx-FT | 86.06 | 7.89 | 74.60 | 58.72 | 37.58 | 4.17 | 87.87 | 4.02 | 78.80 |
| pFedMe | 86.50 | 8.52 | 75.00 | 65.08 | 26.59 | 27.75 | 86.35 | 4.43 | 78.76 |
| pFedMe-FT | 87.06 | 8.02 | 75.00 | 74.36 | 27.02 | 32.49 | 85.47 | **3.06** | 80.95 |
| HypCluster | 82.34 | 9.72 | 68.57 | 57.29 | 39.27 | 0.00 | 86.82 | 5.06 | 77.66 |
| HypCluster-FT | 86.58 | 7.84 | 75.71 | 56.47 | 42.59 | 0.00 | 85.97 | 5.69 | 76.45 |
| FedBN | 85.38 | 8.19 | 74.26 | 76.30 | 22.02 | 44.85 | 87.97 | 3.42 | 81.77 |
| FedBN-FT | 87.65 | **6.33** | **80.02** | 68.50 | 26.83 | 29.17 | 87.02 | 3.47 | 80.13 |
| FedBN-FedOPT | 87.27 | 7.34 | 76.87 | 65.59 | 31.07 | 22.22 | 87.43 | 4.64 | 80.81 |
| FedBN-FedOPT-FT | 87.13 | 7.36 | 78.27 | 68.42 | 28.18 | 30.71 | 87.02 | 3.94 | 81.78 |
| Ditto | 87.18 | 7.52 | 78.23 | 49.94 | 40.81 | 0.00 | 86.85 | 3.98 | 80.44 |
| Ditto-FT | 84.30 | 8.16 | 73.95 | 54.34 | 39.26 | 0.00 | 87.10 | 3.52 | 80.46 |
| Ditto-FedBN | **87.82** | 7.19 | 77.78 | 49.44 | 41.80 | 0.00 | 87.75 | 3.70 | 81.82 |
| Ditto-FedBN-FT | 85.16 | 7.98 | 75.25 | 52.18 | 39.85 | 0.00 | 87.43 | 3.77 | 81.15 |
| Ditto-FedBN-FedOpt | 87.64 | 7.08 | 78.23 | 55.61 | 40.43 | 1.39 | 87.27 | 3.90 | 79.14 |
| Ditto-FedBN-FedOpt-FT | 85.71 | 7.91 | 75.81 | 53.16 | 34.75 | 9.72 | 87.10 | 3.79 | 80.93 |
| FedEM | 82.61 | 9.57 | 69.29 | **76.53** | 23.34 | 44.44 | 85.05 | 4.44 | 78.51 |
| FedEM-FT | 84.91 | 8.39 | 73.64 | 64.29 | 32.84 | 12.96 | 85.54 | 4.48 | 79.39 |
| FedEM-FedBN | 82.94 | 9.35 | 70.43 | 75.06 | **18.48** | **53.33** | 87.63 | 4.14 | **82.54** |
| FedEM-FedBN-FT | 87.09 | 9.24 | 76.36 | 64.33 | 35.72 | 8.59 | 85.68 | 4.33 | 79.44 |
| FedEM-FedBN-FedOPT | 80.48 | 11.02 | 64.84 | 72.66 | 27.18 | 34.17 | 87.11 | 4.24 | 80.32 |
| FedEM-FedBN-FedOPT-FT | 86.23 | 8.33 | 75.44 | 58.42 | 31.21 | 17.93 | 87.16 | 3.66 | 82.20 |

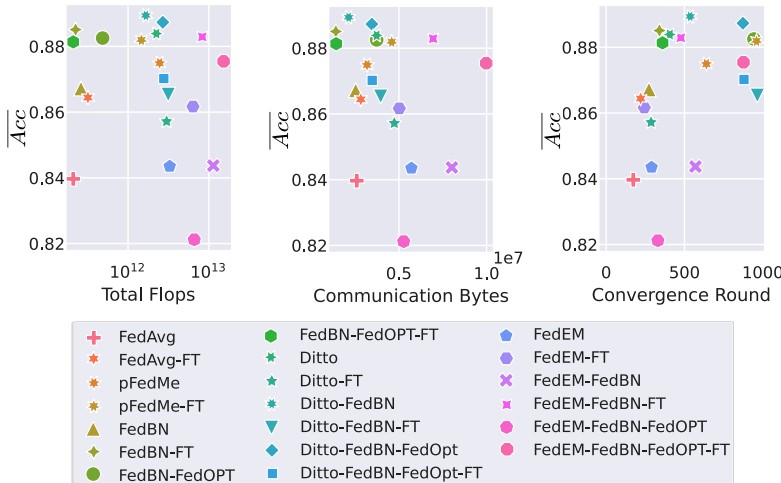

Figure 5: The trade-off between accuracy and efficiency metrics on FEMNIST dataset with client sampling rate $s = 0.2$. The pFL methods usually incur much larger computation and communication costs than non-personalized methods. Full results for other datasets are shown in Appendix D.3.

## 5.3 Efficiency

To quantify the systematical payloads of personalization that is introduced into the FL process, we count the total FLOPs, communication bytes and convergence rounds to demonstrate the trade-off between these metrics and accuracy in Figure 5. Not surprisingly, pFL methods usually incur much larger computation and communication costs than non-personalized methods, requiring more careful and efficient design for further pFL research in resource-limited scenarios. Another interesting observation is that when combined with FT or FedOpt that aggregates the clients' model updates as pseudo-gradients for the global model, the convergence speeds are improved for some methods such as FedEM and Ditto, showing the potential of co-optimizing from the server and local client sides.

## 5.4 More Experiments

Due to the space limitation, we present more experimental results in Appendix D, including results in terms of generalization (Sec.D.1), fairness (Sec.D.2) and efficiency (Sec.D.3) for all the datasets in Table 1. To demonstrate the potential and ease of extensibility of the pFL-bench, we also conduct experiments in the scenario of heterogeneous device resource based on FedScale [38] in Sec.D.4, where we adopt the over-selection mechanism for server and a temporal event simulator [89]. The simulator executes the behaviors of clients according to the virtual timestamps of their message delivery to the server, and the virtual timestamps are updated by the estimated execution time based on different clients' computational and communication capacities. This enables us to simulate different response speeds and participating degrees of clients, which correspond to heterogeneous real-world mobile devices. Moreover, in Sec.D.5, we show that pFL-Bench supports the exploration of trade-offs between pFL and privacy protection techniques, and conduct demonstrative experiments with Differential Privacy [90].

## 6 Conclusions

In this paper, we propose a comprehensive, standardized, and extensible benchmark for personalized Federated Learning (pFL), pFL-Bench, which contains 12 dataset variants with a wide range of domains and unified partitions, and more than 20 pFL methods with pluggable and easy-to-extend pFL subroutines. We conduct extensive and systematic comparisons and conclude that designing effective, efficient and robust pFL methods with good generalization and convergence still remains challenging. We release pFL-Bench with guaranteed maintenance for the community, and believe that it will benefit reproducible, easy and generalizable pFL research and potential applications. We also welcome contributions of new pFL methods and datasets to keep pFL-Bench up-to-date.

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
