# OpenReview forum: "pFL-Bench: A Comprehensive Benchmark for Personalized Federated Learning"
_NeurIPS.cc/2022/Track/Datasets_and_Benchmarks — NeurIPS 2022 Datasets and Benchmarks _

### Official Review · Reviewer_kEb3 · 2022-07-25
**A comprehensive benchmark on personalized federated learning**

**Rating:** 6
**Confidence:** 4

**Strengths:**

1. This paper introduced the first unified experimental setup for personalized federated learning. This is critical since existing papers might use different settings which makes it hard to compare the results fairly with each other.
2. The number of datasets and number of methods used in the benchmark are comprehensive, and evaluation criterions could provide a more holistic view of each method.


**Weaknesses:**

The dataset and evaluation setup seems to be missing the following description:
1. For setup such as fine-tuning, there needs to be a train/test split for local data within each client, how is this splitting done?
2. Are the train/val/test partitioned by clients, or partitioned within each client, or partitioned randomly as if the dataset is centralized?
3. More in-depth explanation for the accuracy definition in table 2 and 3. For example, what is bottom individual accuracy? I could not find a clear definition in the paper.

For text domain data, it would be nice if there are results on the datasets with a larger number of clients such as StackOverflow or Reddit Comments to understand the personalized approach on larger datasets.


**Additional Feedback:**

Questions:
1. Why subsample 5% of FEMNIST in Table 1?
2. What does s = 2 refer to in Table 2 and 3 for FEMNIST?
3. For SST numbers in Table 2, why does the accuracy of un-participated clients can sometimes be higher than that of participating clients?


**Clarity:**

This paper is well-written and organized for the most part. I have raised what else could be clarified in the weakness section.


**Correctness:**

I believe the methodology used for benchmarking personalized federated learning is correct but more details about the dataset split need to be clarified.


**Documentation:**

This paper included documentation in the repository for running the benchmark.


**Relation To Prior Work:**

This paper clearly stated its difference to prior work.


**Summary And Contributions:**

This paper introduced a benchmark on personalized federated learning with an unified experiment setup on existing commonly used datasets. Specifically, this paper took 11 datasets covering image, text, graph and recommendation domains, and benchmarked 4 different personalized setups. The benchmark included evaluation on generalization on trained/unseen participants, fairness between participants, as well as training efficiency for different personalization methods.

---

> ### Author Response · Authors · 2022-08-26
> **Responses to Reviewer kEb3, Part 2**
>
>
> > Q2: What does s refer to in Table 2 and 3 for FEMNIST?
>
> The $s$=0.2 refer to the client sampling rate sampling, i.e., how many clients will be sampled to train and upload their local models for federated aggregation. This is used to simulate the clients' availability, and some clients may lost connection just as they are not sampled.
>
> > Q3: For SST numbers in Table 2, why does the accuracy of un-participated clients can sometimes be higher than that of participating clients?
>
> Thanks for your insightful question! We investigate the distribution of the number of labels for each client in the SST2 dataset and list them in Table below, where the positive samples have a larger number and are more Non-IID than the negative samples.
>
> | label type | mean   | variance   | min  | max      | skewness | kurtosis |
> |------------|--------|------------|------|----------|----------|----------|
> | 1          | 336.40 | 687,393.58 | 1.00 | 6,355.00 | 4.69     | 26.40    |
> | 0          | 228.85 | 245,677.32 | 1.00 | 3,259.00 | 3.45     | 13.57    |
>
> We also calculate the pair-wise similarity of the client's label distribution in terms of Jensen-Shannon distance. The smaller the Jensen-Shannon distance, the more similar the compared distributions.
> We present the results calculated over all or unseen clients only in Table below, where we find that their differences are generally smaller than the differences calculated for the other datasets.
> Besides, recall that we used the pre-trained BERT-Tiny model as the initialized global model on this dataset, unlike most of the other datasets which were randomly initialized.
> Taking these observations together, we suspect that this phenomenon may be due to the fact that this data is more likely to introduce an inductive bias towards positive samples, making the chance of such better accuracy greater than in other datasets.
>
> | clients type | mean | variance | min  | max  | skewness | kurtosis |
> |--------------|------|----------|------|------|----------|----------|
> | all          | 0.33 | 0.05     | 0.00 | 0.83 | 0.37     | -0.90    |
> | un-seen      | 0.38 | 0.06     | 0.00 | 0.75 | 0.08     | -1.42    |

---

> ### Author Response · Authors · 2022-08-26
> **Responses to Reviewer kEb3, Part 1**
>
> Many thanks for your appreciation and constructive review! We make the following responses point by point to address your comments:
>
> > W1: The dataset and evaluation setup seems to be missing the following description
>
> Many thanks for your questions! We have made these description more clear in our revision.
> - W1.Q1 & W1.Q2: for the data splitting, we adopt a unified manner for all datasets by splitting the train/val/test sets *within each client*. That is, for the datasets naturally partitioned by clients (e.g., FEMNIST, each client has data for a writer), we random split train/val/test sets for each client using the ratio reported in Appendix; For the datasets partitioned by LDA, we firstly split the whole data into different parts for different clients, then random split train/val/test sets for each client using the ratio reported in Appendix.
> - W1.Q3: for the bottom individual accuracy, denote the number of all evaluated clients as $N$, the bottom individual accuracy indicates the $\lfloor N/10 \rfloor$)-th worst accuracy. To align with FedEM, the 90th percentile is considered here to omit the particularly noisy results from clients with worse performance with very small data size.
>
>
> > W2 & Q1: Lack larger text datasets and why subsample 5% of FEMNIST
>
> Thank you very much for your comments on the adopted datasets! In addition to the two recommendation datasets already used at the thousand client scale, our benchmark indeed supports other larger datasets thanks to the good integration and extendibility based on FederatedScope.
> - It is worth noting that simulation with pFL algorithms on a large client scale is very challenging, due to the fact that **we need to maintain distinct (personalized) model object for each client**. Let's take the famous benchmark FEMNIST as an example, which has 3,550 users and suppose we adopt the widely-used two-layer CNN network. Although this model only occupies ~200MB, maintaining 3,550 such models would consume more than ~700GB memory.
> Different from non-personalized FL algorithms, for which it is feasible to maintain only one model object for all the clients, for pFL algorithms, we may have to switch and cache the personalized models among CPU, GPU and even disks. Due to the large number of baselines and datasets included in our benchmark, and the corresponding huge hyper-parameter search space, we used several subsets of the FL datasets to reduce the reproduction and experimental barriers.
> - To further enrich considered datasets, we newly conduct experiments on the **Twitter** dataset from LEAF for sentiment analysis task. We adopt a subset which contains **13,203** users, the partition manner for this dataset is natural w.r.t. users rather than by LDA, and the median number of data samples per client is 7. We include a more detailed introduction and visualization in our revision. Here we list experimental results below and report the full and more readable results in the revision pdf. Interestingly, on this highly Non-IID dataset, the *Isolated* method and fine-tuning technique show good performance since users have very small local data sizes and diverse textual expression patterns. Besides, the average accuracy results weighted by local data size ($\overline{Acc}$) are larger than the uniform averaged ones ($\overline{Acc}'$), and the bottom accuracy ($\breve{Acc}$) is 0 in most cases, indicating that the clients having small numbers of local data still have large room to improve their performance.
>
> |  | $\overline{Acc}$ | $\widetilde{Acc}$ | $\Delta$ | $\overline{Acc}'$ | $\sigma$ | $\breve{Acc}$ | FLOPs | Com. | $T'$ |
> |---|---|---|---|---|---|---|---|---|---|
> | Global-Train | 55.56 | - | - | 55.56 | - | - | 31.19G | - | 20.33 |
> | Isolated | 70.04 | - | - | 67.44 | 37.86 | 0 | 36.3M | - | 0 |
> | FedAvg | 62.15 | 61.24 | -0.91 | 56.98 | 37.97 | 0 | 726.45K | 60.32K | 223.33 |
> | FedAvg-FT | 70.53 | 71.17 | 0.64 | 68.45 | 36.19 | 0 | 4.75M | 19.26K | 70.67 |
> | FedOpt | 62.09 | 61.64 | -0.45 | 59.95 | 37.88 | 0 | 726.45K | 60.32K | 220.33 |
> | FedOpt-FT | 71.08 | 71.41 | 0.33 | 69.2 | 36.28 | 0 | 4.52M | 18.35K | 66.67 |
> | pFedMe | 63.45 | 62.52 | -0.94 | 58.18 | 36.67 | 0 | 603.46K | 29.3K | 106.67 |
> | pFedMe-FT | 84 | 71.8 | -12.2 | 78.82 | 33.28 | 22.22 | 12.07M | 15.57K | 53.33 |
> | Ditto | 70.23 | 49.6 | -20.63 | 66.9 | 38.05 | 0 | 2.64M | 38.42K | 140.33 |
> | Ditto-FT | 69.99 | 51.32 | -18.67 | 66.91 | 38.08 | 0 | 4.2M | 36.59K | 133.33 |
> | FedEM | 63.44 | 62.68 | -0.75 | 61.7 | 37.72 | 0 | 18.82M | 95.64K | 163.33 |
> | FedEM-FT | 70.97 | 71.59 | 0.62 | 70.19 | 36.35 | 0 | 24.19M | 35.27K | 40.57 |

---

### Official Review · Reviewer_9wJe · 2022-07-26
**A benchmark system for personalized federated learning**

**Rating:** 6
**Confidence:** 4
**Clarity:** The paper is well written.

**Strengths:**

Propose a new evaluation protocol and conduct experiments in great detail. Report performance of different pFL algorithms in the aspect of generalization, fairness, and system efficiency.

**Weaknesses:**

1. Lack of comparison with the existing benchmark system. How hard is it for other frameworks (Flower, FedML, FedScale) to adapt this new evaluation protocol? Do they provide APIs to do so? Is there any key system design difference that makes pFL not be supported by other frameworks?

2. Lack of analysis and comparison of existing pFL evaluation protocols. What are these protocols and what new aspects does your new unified evaluation protocol bring to us. Unlike the general FL optimization, pFL optimization can be diverse, and so are the evaluation metrics. I didn’t see discussions on different evaluation metrics on pFL. Can your evaluation protocol cover all these parts? If not, why not provide flexible evaluation APIs to support a comprehensive evaluation.

3. Lack of insights and takeaways from evaluation results. I did not see an in-depth analysis of the experiments. Which pFL algorithm is better at different aspects under different scenarios. Which aspects are largely ignored or misunderstood by pFL evaluations before and what are the insights for future pFL based on your results.

4. Other aspects: 1. Evaluations on system efficiency are not convincing, since the clients' heterogeneous system dynamics are not considered currently. 2 I notice the dataset and model is relatively small scale compared to other FL benchmarks. 3. Communication bytes and convergence round is insufficient to picture the system cost in practical FL. These can’t tell much without the device characteristic and network condition.





**Additional Feedback:**

Please refer to the sections above.

**Correctness:**

The results seem largely correct. The choice of evaluation methods is appropriate, which is a strength of the paper. However, I expect to see more discussion on the benchmark results.

**Documentation:**

Yes. The code base is open source.

**Ethics:**

No.

**Relation To Prior Work:**

As I discussed in the weakness part.

**Summary And Contributions:**

In this paper, they propose a new benchmark system pFL-Bench for personalized federated learning. pFL-Bench contains 11 datasets and 20 pFL baselines. They make the following claims about the existing FL benchmark systems

1. They mostly benchmarked on general FL algorithms

2. None of them support the new clients generalization evaluation

3. Few existing benchmarks simultaneously support comprehensive evaluation for trade-offs among accuracy, fairness, and systematical costs

This paper fixes those above by making the following contributions
1. A modular and easy-to-extend pFL codebase with more than 20 competitive pFL baseline implementations

2. Systematic evaluations under containerized environments in terms of generalization, fairness, system overhead, and convergence.

---

> ### Author Response · Authors · 2022-08-26
> **Responses to Reviewer 9wJe, Part 3/3**
>
> > The clients' heterogeneous system dynamics are not considered
>
> Many thanks for your comments! We are glad that the proposed pFL-Bench has good extensibility to support experiments in heterogeneous device resource scenarios, where clients have different computational and communication capacities.
> - Within the response period, we **newly integrate FedScale [1] into our benchmark** with a simulator that executes the behaviors of clients according to virtual timestamps of their message delivery to the server, **leading to heterogeneous system dynamics**.
> - The virtual timestamps are updated by the estimated execution time based on clients' computational and communication capacities with the cost model proposed in FedScale.
> - The server employs an over-selection mechanism for clients at each broadcast round, and thus some clients' messages may be dropped since the clients have different system capacities and different response speeds corresponding to [real-world mobile devices](https://github.com/SymbioticLab/FedScale/tree/master/benchmark/dataset/data/device\_info).
>
> Here we take the Ditto method on FEMNIST dataset as an example and present the results of experiments with heterogeneous device resources below and in our revision.
> - Let $s$ be the clients sampling rate for each FL round, and $s_{agg}$ be the minimal ratio of received feedback w.r.t. the number of clients for the server to trigger federated aggregation in over-selection mode.
> For the homo-device case, we set $s=0.2$ and for the hetero-device case, we set the $s=0.25$ and $s_{agg}=0.8$, leading to the same number of clients used for each federated aggregation.
> - From the results, we can see that the hetero-device version has slower convergence speeds ($T'=0$ indicates that the early-stopping is not triggered within the large number of FL rounds $T=1000$), and it gains worse performance than the homo-device version, especially for the bottom accuracy ($\breve{Acc}$) and standard deviation of the average accuracy ($\sigma$).
> This shows unfairness among clients due to the fact that some low-resourced clients have too long computation or communication time to make their feedback incorporated into the federated aggregation, calling for more considerations w.r.t. device heterogeneity within pFL algorithm design.
>
> || $\overline{Acc}$ | $\tilde{Acc}$ | $\Delta$ | $\overline{Acc}'$ | $\sigma$ | $\breve{Acc}$ | FLOPS | Com. | $T'$ |
> |--|--|--|--|--|--|--|--|--|--|
> | Ditto, Homo-Device | 88.39 | 2.2 | -86.19 | 87.18 | 7.52 | 78.23 | 849.3G | 2.81M | 610 |
> | Ditto, Hetero-Device | 79.76 | 1.43 | -78.33 | 77.39 | 11.25 | 61.76 | 1.72T | 5.72M | 0 |
>
> > More metrics to picture the system cost
>
> Many thanks for your comments! Besides the FLOPS, communication bytes and convergence round, our benchmark indeed supports monitoring more runtime metrics. Thanks to the good integration with wandb, we have already monitored the usage of system resources in runtime, including utilization of CPU, GPU, memory, disk, etc.
>
> In the below table and revision (Appendix Table 21), we newly report the average and peak process memory usage (in MB) and process running times (in seconds). In general, most pFL algorithms do have higher time and space overheads.
> We omit to report results for other metrics since we started very many sets of experiments concurrently, taking up as much of the graphics card's memory and maximizing CPU/GPU utilization as possible, these metrics do not differ much from one of our different experiments. However, it is worth noting that they can be used to analyze algorithm efficiency, and system performance bottlenecks, and to optimize running speed in single-experiment scenarios.
>
> ||FEMNIST|||SST-2|||
> |---|---|---|---|---|---|---|
> ||MEM_avg|MEM_peak|T_run|MEM_avg|MEM_peak|T_run|
> |Global-Train|87|87|11|87|93|892|
> |Isolated|747|1352|1108|118|151|994|
> |FedAvg|10507|17707|840|297|400|319|
> |FedAvg-FT|13400|21752|1193|314|372|545|
> |FedProx|19389|20729|1415|6635|7378|672|
> |FedProx-FT|21209|22504|1935|7805|8161|1810|
> |pFedMe|1881|1980|7205|263|366|1572|
> |pFedMe-FT|18574|21332|4676|282|368|1080|
> |HypCluster|21660|22387|2803|6942|7510|678|
> |HypCluster-FT|22569|23589|3602|7624|8233|887|
> |FedBN|11135|15407|1011|280|373|332|
> |FedBN-FT|17347|25650|936|267|332|551|
> |FedBN-FedOPT|21867|35335|2881|216|311|202|
> |FedBN-FedOPT-FT|9827|14711|845|261|350|283|
> |Ditto|1128|1268|4628|149|205|638|
> |Ditto-FT|1111|1533|8915|229|292|599|
> |Ditto-FedBN|1116|1227|4049|197|229|593|
> |Ditto-FedBN-FT|1454|1730|8528|216|290|627|
> |Ditto-FedBN-FedOpt|1177|1274|5782|203|238|560|
> |Ditto-FedBN-FedOpt-FT|1299|1565|7545|275|334|637|
> |FedEM|940|1063|5070|267|358|1568|
> |FedEM-FT|564|619|7592|327|374|3210|
> |FedEM-FedBN|572|912|12051|270|337|1615|
> |FedEM-FedBN-FT|490|617|9391|307|368|3040|
> |FedEM-FedBN-FedOPT|756|842|8977|251|343|1314|
> |FedEM-FedBN-FedOPT-FT|607|639|19137|301|353|3116|
>
> **References**
>
> [1] FedScale: Benchmarking Model and System Performance of Federated Learning at Scale. ICML 2022.

---

> ### Author Response · Authors · 2022-08-26
> **Responses to Reviewer 9wJe, Part 2/3**
>
> > Insights from evaluation results
>
> Thanks for your question! With the evaluation results, we pointed out the superiority and weaknesses of compared baselines with a large number of datasets, evaluation scenarios, and evaluation metrics. Specifically, we compare them in terms of generalization (Sec. 5.1, with varied datasets, the Non-IID degrees, and the client sampling rates), fairness (Sec. 5.2) and efficiency (Sec. 5.3).
>
> In a nutshell, we found that pFL methods indeed gain better intra-client generalization in many scenarios. However, there is still plenty of room for further pFL research such as enhancing generalization on un-participated clients, being robust on extreme Non-IID cases, computational and communication efficiency, and designing for domain- and model-specific pFL techniques. We believe there are many points worth exploring in depth in these extensive results, and as an early pFL benchmark, we would like to provide the pFL community with a comprehensive and extensible evaluation platform and tool, as well as point out some clues and leave more in-depth research to future pFL work.
>
>
> > Dataset is relatively small
>
> Thank you very much for your comments on the adopted datasets! In addition to the two recommendation datasets already used at the thousand client scale, our benchmark indeed supports other larger datasets thanks to the good integration and extendibility based on FederatedScope.
> - It is worth noting that simulation with pFL algorithms on a large client scale is very challenging, due to the fact that **we need to maintain distinct (personalized) model object for each client**. Let's take the famous benchmark FEMNIST as an example, which has 3,550 users and suppose we adopt the widely-used two-layer CNN network. Although this model only occupies ~200MB, maintaining 3,550 such models would consume more than ~700GB memory.
> Different from non-personalized FL algorithms, for which it is feasible to maintain only one model object for all the clients, for pFL algorithms, we may have to switch and cache the personalized models among CPU, GPU and even disks. Due to the large number of baselines and datasets included in our benchmark, and the corresponding huge hyper-parameter search space, we used several subsets of the FL datasets to reduce the reproduction and experimental barriers.
> - To further enrich considered datasets, we newly conduct experiments on the **Twitter** dataset from LEAF for sentiment analysis task. We adopt a subset which contains **13,203** users, the partition manner for this dataset is natural w.r.t. users rather than by LDA, and the median number of data samples per client is 7. We include a more detailed introduction and visualization in our revision. Here we list experimental results below and report the full and more readable results in the revision pdf. Interestingly, on this highly Non-IID dataset, the *Isolated* method and fine-tuning technique show good performance since users have very small local data sizes and diverse textual expression patterns. Besides, the average accuracy results weighted by local data size ($\overline{Acc}$) are larger than the uniform averaged ones ($\overline{Acc}'$), and the bottom accuracy ($\breve{Acc}$) is 0 in most cases, indicating that the clients having small numbers of local data still have large room to improve their performance.
>
> |  | $\overline{Acc}$ | $\widetilde{Acc}$ | $\Delta$ | $\overline{Acc}'$ | $\sigma$ | $\breve{Acc}$ | FLOPs | Com. | $T'$ |
> |---|---|---|---|---|---|---|---|---|---|
> | Global-Train | 55.56 | - | - | 55.56 | - | - | 31.19G | - | 20.33 |
> | Isolated | 70.04 | - | - | 67.44 | 37.86 | 0 | 36.3M | - | 0 |
> | FedAvg | 62.15 | 61.24 | -0.91 | 56.98 | 37.97 | 0 | 726.45K | 60.32K | 223.33 |
> | FedAvg-FT | 70.53 | 71.17 | 0.64 | 68.45 | 36.19 | 0 | 4.75M | 19.26K | 70.67 |
> | FedOpt | 62.09 | 61.64 | -0.45 | 59.95 | 37.88 | 0 | 726.45K | 60.32K | 220.33 |
> | FedOpt-FT | 71.08 | 71.41 | 0.33 | 69.2 | 36.28 | 0 | 4.52M | 18.35K | 66.67 |
> | pFedMe | 63.45 | 62.52 | -0.94 | 58.18 | 36.67 | 0 | 603.46K | 29.3K | 106.67 |
> | pFedMe-FT | 84 | 71.8 | -12.2 | 78.82 | 33.28 | 22.22 | 12.07M | 15.57K | 53.33 |
> | Ditto | 70.23 | 49.6 | -20.63 | 66.9 | 38.05 | 0 | 2.64M | 38.42K | 140.33 |
> | Ditto-FT | 69.99 | 51.32 | -18.67 | 66.91 | 38.08 | 0 | 4.2M | 36.59K | 133.33 |
> | FedEM | 63.44 | 62.68 | -0.75 | 61.7 | 37.72 | 0 | 18.82M | 95.64K | 163.33 |
> | FedEM-FT | 70.97 | 71.59 | 0.62 | 70.19 | 36.35 | 0 | 24.19M | 35.27K | 40.57 |

---

> ### Author Response · Authors · 2022-08-26
> **Responses to Reviewer 9wJe, Part 1/3**
>
> Thank you very much for your appreciation and constructive and detailed review! We make the following responses point by point to address your comments:
>
> > Lack of comparison to existing FL bench, and how easy it integrate with other FL frameworks
>
> Many thanks for your comments!
> Regards the comparison to existing FL bench, we have discussed the difference between ours and theirs in Sec. 2 "Related Work".
> - To summarize, most existing FL benchmarks contain general FL algorithms, lacking recently proposed pFL methods that are considered in our benchmark. For example, LEAF includes FedAvg and Mocha; FedScale includes FedAvg, FedProx, and FedYoGi; Flower includes FedAvg, FedProx, QFedAvg and FedOptim.
> - Moreover, we support the new clients generalization evaluation and simultaneously support comprehensive evaluation for trade-offs among generalization, fairness and system costs.
>
> Regards the integration with other FL frameworks,
> - to demonstrate the extendibility, we newly integrate FedScale [1] into our benchmark via a simulator that executes the behaviors of clients according to virtual timestamps of clients' messages delivery to the server, such that we can evaluate performance with heterogeneous system dynamics and heterogeneous device information used by FedScale. More details are given in the next response to *"[Part 3] The clients' heterogeneous system dynamics are not considered"*.
> - We would like to clarify that although the specific APIs of our codebase and other FL framework differ, our codebase and many other FL frameworks are both designed to be as modular as possible, and moreover, share some similarities in the flexible, event-driven programming philosophy to describing federation behaviors, such as FedScale and FedML. To further reduce the barriers to use and learning costs of our codebase, we also provide extensive documentation including [APIs](https://federatedscope.io/refs/index), [executable scripts](https://github.com/alibaba/FederatedScope/tree/master/scripts), customized [configurations](https://github.com/alibaba/FederatedScope/tree/master/federatedscope/core/configs) and [tutorials](https://federatedscope.io/docs/quick-start/).
>
>
> > Comparison of existing pFL evaluation protocols and support of comprehensive evaluation
>
> Thank you for your comments!
> Existing pFL works adopt very diverse evaluation protocols, since different pFL algorithms have their own pros and cons, and are suitable for a variety of applications just as our benchmark results show. Different methods may emphasize more their favorable results/setups where it shines. For example, most pFL works focus on improving accuracy while neglecting fairness (e.g., at the expense of some clients with fewer data, as those clients are given less weight when their parameters are used in federated aggregation), and system costs (e.g., meta-learning based methods and multi-model based methods usually introduce a lot of additional computational and communication overhead). Besides, a few pFL works consider the generalization on FL-non-participated clients.
>
> Without a comprehensive benchmark, *it is difficult for the community to evaluate a new approach from different metrics, scenarios and datasets at the same time*. We would like to start by building an end-2-end evaluation protocol that is as comprehensive, easy to use and extensible as possible to make further research easier and more objective.
>
> Specifically, for each experimental run, we track all these considered metrics at the same time by default, including server-side and clients-sides monitoring w.r.t. generalization performance (accuracy, loss, etc.,), fairness (std, quantiles, etc.,), and system costs (FLOPS, network traffics, memory, CPU/GPU utilization, etc.,). These metrics to be monitored can be flexibly adjusted through our unified [configuration](https://github.com/alibaba/FederatedScope/tree/master/federatedscope/core/configs#evaluation) and [APIs](https://github.com/alibaba/FederatedScope/tree/master/federatedscope/core/monitors). To make this part clearer, we have also added an sub-section in the appendix to summarise these metrics with more details (Sec. B.2).

---

> > ### Comment · Reviewer_9wJe · 2022-08-29
> > **Thank you for the response.**
> >
> > I appreciate your effort in addressing my questions. I think most of my concerns have been resolved. I will raise my score from 5 to 6.

---

### Official Review · Reviewer_CoEZ · 2022-07-27
**Good paper, sigificant contribution to the field of personalized federated learning**

**Rating:** 7
**Confidence:** 4

**Strengths:**

The paper is well motivated and helps to overcome the absence of a standard benchmark for personalized federated learning. The paper proposes 11 datasets with different tasks, sizes, and levels of heterogeneity, and conducts extensive experiments to evaluate a reasonable amount of "standard" and personalized FL algorithms in terms of test accuracy, fairness and efficiency. To the best of my knowledge, this work is the most complete benchmark in the field of personalized federated learning.

The authors made a significant effort to include all the aspects of personalized federated learning in their comparison. They provide weighted average accuracy on seen and unseen clients, as long as the equally-weighted accuracy, the bottom decile accuracy and the standard deviation of the average accuracy. The authors also provide results to quantify the effect of heterogeneity on different methods, and give an estimation of the computation, memory and communication cost of the of the algorithms they study.

The authors also propose a clear plan to maintain the benchmark and to extend to include additional aspect of personalized federated learning.


**Weaknesses:**

Despite the effort of the authors and the overall relevance of the paper, I think it still need some improvement.

First, the generalized FL formulation discussed in Section 3.1 is not totally accurate; for example, the formulation in Eq. (1) does not capture Ditto [35]. The true objective of personalized FL methods is to solve for each client $i$ the following optimization problem
$$\min_{\theta_{i}} {\mathbb{E}}_{\mathcal{D}_i}[f(\theta_i; x, y)],$$
corresponding to the selection of the parameters that minimize the true risk of client $i$. Of course client $i$ can not minimize its true risk, because he does not have access to the local data distribution $D_i$, instead client $i$ only has access to a finite, usually small, number of samples drawn from $D_i$. Most papers, considers the minimization of an empirical risk computed using the aggregated dataset and the local dataset of a client, with some reguralization on the distance between the global and local models, see for example Eq. (1) in (Hanzely et al., 2021). I highly recommend the authors to update Section 3.1, good references include (Mansour et al, 2020) and (Mohri et al, 2019).

Second, some experimental results are unexpected, I do not understand why FedAvg outperforms (by a large margin) Global-Train on FEMNIST for example. I think that the implementation/execution of Global-Train need to be double checked.

Third, the benchmark is based on FederatedScope, which creates an entry barrier to the usage of the benchmark as it requires first learning how to use FederatedScope. The benchmark will be used and adopted in more papers, if the code was provided in native PyTorch or TensorFlow because they are a lot more popular. However, I understand the choice of the authors to use FederatedScope.


**Additional Feedback:**

I want to thank the authors for their efforts. This paper is valuable for the personalized FL community and I recommend acceptance. I strongly recommend the authors to updated Section 3.1 and to double check the implementation of Global-Train.

I encourage the authors to include other methods/datasets in the comparison, and suggest to add the results of this study to [paperswithcode](https://paperswithcode.com/task/personalized-federated-learning).

## References

Hanzely, Filip, Boxin Zhao, and Mladen Kolar. "Personalized federated learning: A unified framework and universal optimization techniques." arXiv preprint arXiv:2102.09743 (2021).

Mohri, Mehryar, Gary Sivek, and Ananda Theertha Suresh. "Agnostic federated learning." In International Conference on Machine Learning, pp. 4615-4625. PMLR, 2019.

**Clarity:**

The paper is clear and easy to follow. The writing quality is acceptable but need to be improved, and the paper has some typos.

Line 93: that are perform well -> the perform well

Line 109: that do not participate the FL training -> that do not participate to the FL training

Line 118: that are not contribute -> that do not contribute

**Correctness:**

The paper is overall correct. I have few comments on some claims that I included in Section Weaknesses.

**Documentation:**

The documentation is okay.

**Relation To Prior Work:**

The paper has a decent  related work discussion.

**Summary And Contributions:**

This paper benchmarks "standard" federated learning algorithms and some recent personalized federated learning (pFL) algorithms on 11 federated learning datasets, spanning a wide range of machine learning tasks including image classification, natural language processing, graph classification, and recommender systems. The comparison includes different criteria to measure average accuracy, fairness, and energetic efficiency.

The experimental results demonstrate on one hand that personalization is a promising solution to handle statistical heterogeneity in federated learning, and on the other hand that no method can consistently outperform others across all metrics and datasets.

---

> ### Author Response · Authors · 2022-08-26
> **Responses to Reviewer CoEZ**
>
>
> We sincerely thank you for your appreciation and constructive review! We make the following responses point by point to address your comments:
>
> > Minor issues on Equation (1)
>
> Many thanks for your comments and suggestions! We would like to clarify that the coefficients $\alpha$, $\beta$, and $\gamma$ trade off the different emphasis on these terms. And when $\alpha=0, \beta=1, \gamma=0$, the objective becomes the one for your mentioned Ditto case. Moreover, to make this part more accurate, we modified the $R$ term into $R\big(\{\mathbb{E}\_{(x, y) \sim D\_j} [f(\theta\_j; x, y) |\theta\_g] \}\_{j\in\tilde{C}} \big)$ in our revision, which indicates that the personalization with local model $\theta\_j$ also happens for the unseen clients $\mathcal{\tilde{C}}$ but conditioned on the global model trained on seen clients $\theta\_g$.
>
>
> > Global-Train method performs bad on FEMNIST
>
> Many thanks for your comments! We apologize for the error in the result of the Global-Train method in Table 1, which we have fixed from 52.48 to 74.51 in the revision. After re-running the method with the tracked config file, and double-checking the logs of historical experiment results, we found that the results of the experiment from one of the three runs at that time were wrong and somewhat out of date (before we fixed the bug Global-Train mode in this [pull request](https://github.com/yxdyc/FederatedScope/commit/7bc3eeb4a29d40ca8a4eb111c266ae077fe25759)). For now, we also double-checked the codes for the relevant experiment results processing and the logs for other experimental runs. Again, thank you very much for pointing this out!
>
>
> > Based on FederatedScope, which creates an entry barrier
>
> Many thanks for your comments!
> - Currently, most of the foundational components, built-in algorithms, models, data, etc. we provide are based on pytorch and are ready to use out of the box, making it easy for users familiar with pytorch to get started. In addition, the FederatedScope library provides the ability to support [cross ML backends](https://federatedscope.io/docs/simulation-and-deployment/#cross-ml-backends) and extend TensorFlow codes due to its message-oriented programming paradigm.
> - For those familiar with other FL frameworks, we would like to clarify that although the specific APIs of our codebase and other FL framework differ, our codebase and many other FL frameworks are both designed to be as modular as possible, and moreover, share some similarity in the flexible, event-driven programming philosophy to describing federation behaviors, such as FedScale and FedML. To demonstrate the extendibility, we newly integrate FedScale [1] into our benchmark via a simulator that executes the behaviors of clients according to virtual timestamps of clients' messages delivered to the server, such that we can evaluate performance with heterogeneous system dynamics and heterogeneous device information used by FedScale. More details are given in the response to Reviewer Ytx8 "["Weakness" Part, page 2/5] Lack of support/integration of system heterogeneous FL".
> To further reduce the barriers to use and learning costs of our codebase, we also provide extensive documentation, including [APIs](https://federatedscope.io/refs/index), [executable scripts](https://github.com/alibaba/FederatedScope/tree/master/scripts), customised [configurations](https://github.com/alibaba/FederatedScope/tree/master/federatedscope/core/configs) and [tutorials](https://federatedscope.io/docs/quick-start/). Thanks again for your comments!
>
>
> > A few typos
>
> Thank you for pointing out these typos! We modified them accordingly in the revision and carefully proofread our paper again.
>
>
> **References**
>
> [1] FedScale: Benchmarking Model and System Performance of Federated Learning at Scale. ICML 2022.

---

> > ### Comment · Reviewer_CoEZ · 2022-08-29
> > **Comment on Equation 1**
> >
> > Thank you for answer, and for addressing m concerns. I still do not agree on your comment on Equation 1. With the choice $(\alpha, \beta, \gamma) = (0, 1, 0)$, Problem (1) is reduced to $\min_{\theta_{i}}L(\\{E_{x,y}[f(\theta_{i}; x, y)]\\}_{i\in C})$.
> >
> > I am not able to find a choice of $L$, such that Problem (1) is equivalent to the objective considered in Ditto, i.e.,
> >
> > $$\min_{\theta_{i}} E_{x,y}[f(\theta_{i}; x, y)] + \frac{\lambda}{2} \\| \theta_{i} - \theta^{*}\\|^{2}, $$
> >
> > s.t., $\theta^{*} \in \text{arg}\min_{\theta}G(F_{1}(\theta), \dots, F_{|C|}(\theta))$.
> >
> > Would you please clarify this point.

---

> > > ### Author Response · Authors · 2022-08-29
> > > **Further responses to Reviewer CoEZ**
> > >
> > > Many thanks for your rigorous comments and helpful feedback! After further careful review and discussion of this subsection, we agree that Equation 1 needs some additional modifications.
> > >
> > > We have just uploaded a new revision and added a new term $Q(\theta\_k, \theta\_g) \_{k \in (\mathcal{C} \cup \tilde{\mathcal{C}})}$ to characterize the explicit modeling of the personalized and global models. We have added the following new sentences:
> > > - in line 115, *"$Q(\cdot)$ term indicates the modeling of the relationship between the local and global models, such as the L2 norm to regularize the model parameters in Ditto, $\frac{\lambda}{2}||\theta_k-\theta_g||^2$."*
> > > - in line 120, *"Besides, different pFL methods may flexibly introduce various constraints on this optimization objective, which we have omitted in Eq. (1) for brevity."*
> > >
> > > Thanks again for your appreciation and dedicated effort to help us enhance the paper!

---

### Official Review · Reviewer_Ekaz · 2022-07-27
**An exhaustive bunch of datasets and models for personalised federated learning with a lot of experimental results**

**Rating:** 7
**Confidence:** 3

**Strengths:**

- a comprehensive list of datasets and methods for personalised FL
- a clear description of the methods used to generate split datasets
- a concise and well referenced explanation of the benchmarked methods
- a large and interesting collection of experimental results


**Weaknesses:**

- It would have been have been nice to see more non-personalised FL methods benchmarked.
- The reviewer would have enjoyed more metrics and figures about the heterogeneity of the split datasets (looking at both inputs and labels) that is a key variable for the topic of this article.
- The github repo should be more user-friendly and the code easier to install and run.
- Some experimental results do look surprising.


**Additional Feedback:**

NA

**Clarity:**

The paper is clear and pleasant to read. Between the main article and the supplementary material, the level of detail given is appropriate for this track.

**Correctness:**

The experimental design is sound and rigorous and the evaluation metrics are relevant and informative.

**Documentation:**

Access to data is very user-friendly. The data collection process is clearly explained.

**Ethics:**

No particular concern on the data privacy and licensing front.
The topic of federated learning in general has a clear positive potential societal impact.

**Relation To Prior Work:**

The related works section is well referenced and gives good context.

**Summary And Contributions:**

The authors have clearly invested a lot of time and effort in this very thorough benchmark.
It is the first to focus on personalised federated learning.
This is new although the reviewer doubts the pertinence of having those methods evaluated separately from non-personalised federated learning methods.
The article is pleasant and clear to read. The supplementary material gives useful details about the datasets, methods, results and the implementation.
The reviewer was not able to run any experiment or look at any code as the choice of a docker image makes it cumbersome.
Overall, this is a very comprehensive set of datasets and models that can foster FL innovation and fits nicely with the purpose of the NeurIPS Datasets and Benchmarks track.

---

> ### Author Response · Authors · 2022-08-26
> **Responses to Reviewer Ekaz**
>
> Many thanks for your appreciation and constructive comments! We make the following responses point by point to address your comments:
>
> > It would have been have been nice to see more non-personalised FL methods benchmarked.
>
> Thank you for your comments, our framework does support and can be easily extended to more non-personalized FL methods, thanks to the modular design, the flexible message-oriented programming paradigm, and the already built-in FL algorithms and continuously increasing implementations in the FederatedScope package.
>
> To further demonstrate the extensibility of pFL-Bench, we conduct new experiments with a non-personalized pFL method, **FedProx**[1], which introduces a proximal term to encourage the updated models at clients not to differ too much from the global model.
> We have listed some of the results below and present more complete results in the revision. We observe that FedProx usually achieves worse accuracy than Ditto and pFedMe, which also leverage proximal term to conduct the model parameter regularization during the local learning processes. The main difference is that Ditto and pFedMe maintain client-specific local models instead of training the same global model for different clients, showing the effectiveness of personalization.
>
> |  | FEMNIST, s=0.2 |  |  | SST-2 |  |  | PUBMED |  |  |
> |---|---|---|---|---|---|---|---|---|---|
> |  | $\overline{Acc}$ | $\widetilde{Acc}$ | $\Delta$ | $\overline{Acc}$ | $\widetilde{Acc}$ | $\Delta$ | $\overline{Acc}$ | $\widetilde{Acc}$ | $\Delta$ |
> | FedAvg | 83.97 | 81.97 | -2 | 74.88 | 80.24 | 5.36 | 87.27 | 72.63 | -14.64 |
> | FedAvg-FT | 86.44 | 84.94 | -1.5 | 74.14 | 83.28 | 9.13 | 87.21 | 79.78 | -7.43 |
> | FedProx | 84.1 | 81.49 | -2.61 | 74.36 | 79.2 | 4.84 | 87.23 | 75.02 | -12.21 |
> | FedProx-FT | 87.34 | 85.27 | -2.08 | 79.94 | 80.48 | 0.59 | 88.24 | 79.12 | -9.12 |
> | pFedMe | 87.5 | 82.76 | -4.73 | 71.27 | 69.34 | -1.92 | 86.91 | 71.64 | -15.27 |
> | pFedMe-FT | 88.19 | 82.46 | -5.73 | 75.61 | 66.48 | -9.13 | 85.71 | 77.07 | -8.64 |
> | HypCluster | 83.8 | 81.88 | -1.92 | 46.26 | 61.32 | 15.05 | 87.2 | 75.37 | -11.83 |
> | HypCluster-FT | 87.79 | 85.67 | -2.12 | 52.46 | 78.67 | 26.21 | 86.43 | 76.69 | -9.74 |
> | FedBN | 86.72 | 7.86 | -78.86 | 74.88 | 75.4 | 0.52 | 88.49 | 52.53 | -35.95 |
> | FedBN-FT | 88.51 | 82.87 | -5.64 | 68.81 | 82.43 | 13.63 | 87.45 | 80.36 | -7.09 |
> | Ditto | 88.39 | 2.2 | -86.19 | 52.03 | 46.79 | -5.24 | 87.27 | 2.84 | -84.43 |
> | Ditto-FT | 85.72 | 56.96 | -28.76 | 56.49 | 65.5 | 9.01 | 87.47 | 35.03 | -52.44 |
> | FedEM | 84.35 | 82.81 | -1.54 | 75.78 | 67.67 | -8.11 | 85.64 | 71.12 | -14.52 |
> | FedEM-FT | 86.17 | 85.01 | -1.16 | 64.86 | 81.63 | 16.77 | 85.88 | 78.08 | -7.8 |
>
>
> > The reviewer would have enjoyed more metrics and figures about the heterogeneity of the split datasets.
>
> Thanks for your suggestion! Besides the quantity skew visulization that has been included in our paper, we have currently added a probing tool to calculate clients' pairwise similarity of their feature or label distributions in terms of Jensen–Shannon distance in our codebase. The smaller the Jensen-Shannon distance, the more similar the compared distributions. We list the statistics of calculated Jensen-Shannon distances about label distribution on CIFAR-10 with different LDA $\alpha$ factors below, and plot the histogram in our revision.
>
> We can see that as the degree of heterogeneity increases (the $\alpha$ decreases), the larger the Jensen-Shannon distances we get. We can perform similar calculations on a variety of FL datasets, and further rank these distances, and in turn select those clients whose distributions are very different but whose models do not perform well for further analysis, understanding, and algorithm improvement. Thanks again for your suggestion!
>
> | alpha | mean  | variance | min   | max   | skewness | kurtosis |
> |-------|-------|----------|-------|-------|----------|----------|
> | 5     | 0.389 | 0.007    | 0.099 | 0.722 | 0.028    | -0.086   |
> | 0.5   | 0.525 | 0.01     | 0.164 | 0.833 | -0.178   | -0.064   |
> | 0.1   | 0.669 | 0.014    | 0.056 | 0.833 | -0.825   | 0.375    |
>
>
> > The github repo should be more user-friendly
>
> Many thanks for your comments! To further reduce the barriers to use and learning costs of our codebase, we have recently made a lot of efforts to provide extensive easy-to-access, easy-to-follow installation tutorials and easy-to-read documentation, including various [installing ways](https://github.com/alibaba/FederatedScope#step-1-installation) (Docker, Conda, source), codebase [APIs](https://federatedscope.io/refs/index), [executable scripts](https://github.com/alibaba/FederatedScope/tree/master/scripts), customised [configurations](https://github.com/alibaba/FederatedScope/tree/master/federatedscope/core/configs) and [tutorials](https://federatedscope.io/docs/quick-start/).
>
>
> **References**
>
> [1] Federated Optimization in Heterogeneous Networks. MLSys 2020.

---

### Official Review · Reviewer_Ytx8 · 2022-07-27
**Review for pFL-Bench**

**Rating:** 6
**Confidence:** 3

**Strengths:**

* Personalised FL is a promising research direction of distributed training in the wild that has received attention in the last years. Having a benchmark suite that acts as the common denominator and the golden standard between different techniques can be beneficial for the community.
* The authors have gone the extra mile and implemented various techniques from the literature that can act as initial baselines in the area of personalised FL. Moreover, certain techniques can be combined for further gains. It's also positive that they have integrated datasets representing different modalities.
* I like the brief theoretical background that the authors provide about how personalisation in FL is being pursued in the literature.

**Weaknesses:**

* The paper felt at times more like a survey of personalised FL techniques rather than a benchmarking suite for personalised FL.
* Datasets introduced here are simply gathered from pre-existing literature, and merely federated. This has been done various times before and especially for the selected vision tasks.
    * Some widely used FL datasets are missing, from the NLP (e.g. Reddit, StackOverflow), Speech (e.g. SpeechCommands) or other modalities (e.g. HAR)
* There is lack of support/integration of techniques from the system heterogeneous FL and privacy realms. The tradeoff dynamics between personalisation, privacy and fairness are the most interesting and underexplored in the field.
* The suite is offering implementations based on FederatedScope, which might not be easily "intergratable" with other FL frameworks (e.g. TFF, Flower, FLSIm, etc.)
* The authors argue a lot about the reproducibility of their implementation. However, in cross-device settings, stochasticity may come from client availability (or lack thereof), failures or even asynchronicity of aggregation. It is not clear how PFL-Bench would support that level of determinism/reproducibility.

**Additional Feedback:**

Further support:
* The suite seems to be running everything in a simulated manner. Are there any plans to run across remote workers or on-device? This is particularly important for the case of cross-device setups.
* Will the benchmark support pretrained global models for the users to bootstrap their research?
* It is further unclear if the frameworks supports personalisation per cohort instead of per client.

Limitations of work have very briefly mentioned if at all. I would urge the authors to present limitations of their system more spherically.

**Clarity:**

The paper is generally well-written and easy to follow. However, there are certain presentation issues, laid out below:

* Maybe a more explicit distinction between cross-silo and cross-device settings should be made in the paper and what datasets/techniques correspond to each realm.
* Figure 4: It is very difficult to make out different lines per graph. Maybe the authors should consider an alternative presentation method for this dense information.
* Tables 6-19 should not be inline with the references in the appendix.
* In section 5.3 (some are only made clearer in the relevant tables of the appendix):
    * Are FLOPs counted as the amount of floating operations in training or inference (f/w propagation only)?
    * It is not clear in what granularity are the communication bytes reported. Is it upstream+downstream across users and communication rounds until convegence (with early stopping)?
    * Similarly for the communication rounds, the setting is a bit counter-intuitive. Typically, one either fixes the number of communication rounds or they count them until a certain accuracy threshold is reached or they perform early stopping. It is unclear how the communication rounds were selected per technique here.
* The metrics for fairness could be presented more like a set of commonly accepted benchmark metrics for assessing fairness in FL. Also, fairness in participation could be quantified/included.

In terms of wording, the terms "practicability", "systematical", "un-participated" and "FL course" should be reconsidered.

**Correctness:**

The computational/memory/network costs associated with training have been very coarsely presented and lack definitions and runtime footprints in the main text. The authors are measuring certain proxy performance metrics instead of runtime ones (e.g. FLOPs vs. training latency) and are missing other important ones (e.g. peak memory usage) that would be important from a deployability point of view and especially in the cross-device setting.

Some of the datasets are counted as different while the only thing that changes is the LDA alpha parameter. I would count them uniquely.

The authors start motivating their work by supporting that different approaches in personalised FL are evaluated in different settings and are not directly comparable with one another. However, they reach the conclusion in L229 that there is no dominating technique across the implemented ones (see free lunch theorem).
Given that new work tends to emphasise on the positive results, even under the presence of such a benchmarking suite, the trend of presenting favourable results/setups where a new approach shines  may not be avoided. Thus common baselines/setups may be trickier than offering a benchmarking suite.

Wrt evaluation:
* Why does FEMNIST in the "Global-Train" method perform so poorly?
* Bottom accuracy being zero in table 4 means that it is essentially worse than random.

**Documentation:**

The dataset directs to the master branch of FederatedScope and then only by clicking in a link through the ReadMe is the code implementation accessible. Maybe the authors should consider a separate repository for this benchmarking suite.

I was initially very positive about the existence of dockerfiles for experiment reproducibility. However, upon inspection:
* The Dockerfile link (https://github.com/alibaba/FederatedScope/blob/master/enviroment/docker_files/federatedscope-torch1.8-application.Dockerfile) seems to be inaccessible.
* The Dockerfile from here: https://github.com/alibaba/FederatedScope/blob/Feature/pfl_bench/enviroment/docker_files/federatedscope-torch1.8-application.Dockerfile seems to simply set up the requirements in a container, rather than make any other part of the evaluation more concrete/reproducible. Experiments seem to be running as bash scripts inside that docker container and in a single node. Maybe multinode configurations are supported, but there is no documentation for that.

License is clear (Apache License 2.0). Maintenance will probably be happening as part of FederatedScope (same authors) and contributions seems to be welcome to the project from the community.

**Ethics:**

The paper completely misses the broader societal impact of the work. Examples would include:
* sustainability in training
* privacy issues
* using third-party resources (e.g. clients phones)

The datasets used have already been open, so I do not expect the need for further review on these.

**Relation To Prior Work:**

I see the proposed approach as an attempt to standardise experiments/datasets through a common, already bootstrapped benchmarking suite. However, there have been several FL frameworks with partial support for datasets/personalisation approaches. Maybe a table representing the coverage of techniques/datasets in major FL frameworks would be of great utility to the community.

**Summary And Contributions:**

This paper is introducing a benchmark suite for personalised Federated Learning, with a focus on cross-device settings. It provides 9 pre-split datasets and implements 11 personalised FL baselines methods for these tasks. Implementation is based on FederatedScope.

---

> ### Author Response · Authors · 2022-08-26
> **Responses to Reviewer Ytx8, The remaining Parts, page 5/5**
>
> > Plans to run across remote workers or on-device
>
> Thank you for your question! Actually, we currently have support for running across remote workers, with the distributed mode of FederatedScope (see more details [here](https://federatedscope.io/docs/quick-start/#distributed-mode])). Besides, with FederatedScope, we are exploring personalized FL applications on real-world ARM architecture devices in our internal projects. If things go well, in the future we will consider and be happy to share and dis more practical experiences and support more realistic scenarios with the community.
>
> > Support pretrained models
>
> Thanks for your questions!
> - Our benchmark involved a considerable number of experiments, and most of them were based on global models with random initialization and trained from scratch. For these experiments, starting the FL with a pre-trained global model is another different, but interesting setting from our paper, and we do not provide pre-trained models for them.
> - Another special case is for the experiments involving textual datasets using Transformers, where we have indeed used the pre-trained BERT-Tiny model from huggingface and our codebase will automatically download it according to the configuration "model.tyep=google/bert_uncased_L-2_H-128_A-2@transformers".
>
>
> > "It is further unclear if the frameworks supports personalisation per cohort instead of per client."
>
> Thank you for your comments! Our framework does support and can be easily extended to personalization per cohort, thanks to the modular design, the already built-in multi-model components and the flexible message-oriented programming paradigm. To further alleviate your concern, we have implemented a new clustering-based pFL baseline approach, **HypCluster**[3], which splits clients into clusters, learns different personalized models for different clusters, and we set the number of clusters as 3 in our new experiments.
> We have listed some of the results below and present more complete results in the revision. We can see that the HypCluster achieves slightly worse accuracy than FedEM, which also learns 3 models for each client, while finally uses a stronger mixture ensemble model for inference instead of choosing one of the three clusters to which the client is most likely to belong.
>
> |  | FEMNIST, s=0.2 |  |  | SST-2 |  |  | PUBMED |  |  |
> |---|---|---|---|---|---|---|---|---|---|
> |  | $\overline{Acc}$ | $\widetilde{Acc}$ | $\Delta$ | $\overline{Acc}$ | $\widetilde{Acc}$ | $\Delta$ | $\overline{Acc}$ | $\widetilde{Acc}$ | $\Delta$ |
> | FedAvg | 83.97 | 81.97 | -2 | 74.88 | 80.24 | 5.36 | 87.27 | 72.63 | -14.64 |
> | FedAvg-FT | 86.44 | 84.94 | -1.5 | 74.14 | 83.28 | 9.13 | 87.21 | 79.78 | -7.43 |
> | FedProx | 84.1 | 81.49 | -2.61 | 74.36 | 79.2 | 4.84 | 87.23 | 75.02 | -12.21 |
> | FedProx-FT | 87.34 | 85.27 | -2.08 | 79.94 | 80.48 | 0.59 | 88.24 | 79.12 | -9.12 |
> | pFedMe | 87.5 | 82.76 | -4.73 | 71.27 | 69.34 | -1.92 | 86.91 | 71.64 | -15.27 |
> | pFedMe-FT | 88.19 | 82.46 | -5.73 | 75.61 | 66.48 | -9.13 | 85.71 | 77.07 | -8.64 |
> | HypCluster | 83.8 | 81.88 | -1.92 | 46.26 | 61.32 | 15.05 | 87.2 | 75.37 | -11.83 |
> | HypCluster-FT | 87.79 | 85.67 | -2.12 | 52.46 | 78.67 | 26.21 | 86.43 | 76.69 | -9.74 |
> | FedBN | 86.72 | 7.86 | -78.86 | 74.88 | 75.4 | 0.52 | 88.49 | 52.53 | -35.95 |
> | FedBN-FT | 88.51 | 82.87 | -5.64 | 68.81 | 82.43 | 13.63 | 87.45 | 80.36 | -7.09 |
> | Ditto | 88.39 | 2.2 | -86.19 | 52.03 | 46.79 | -5.24 | 87.27 | 2.84 | -84.43 |
> | Ditto-FT | 85.72 | 56.96 | -28.76 | 56.49 | 65.5 | 9.01 | 87.47 | 35.03 | -52.44 |
> | FedEM | 84.35 | 82.81 | -1.54 | 75.78 | 67.67 | -8.11 | 85.64 | 71.12 | -14.52 |
> | FedEM-FT | 86.17 | 85.01 | -1.16 | 64.86 | 81.63 | 16.77 | 85.88 | 78.08 | -7.8 |
>
> **References**
>
> [1] FedScale: Benchmarking Model and System Performance of Federated Learning at Scale. ICML 2022.
>
> [2] Federated Learning With Differential Privacy: Algorithms and Performance Analysis. IEEE Transactions on Information Forensics and Security 2020.
>
> [3] Three Approaches for Personalization with Applications to Federated Learning. Arxiv 2020.

---

> > ### Comment · Reviewer_Ytx8 · 2022-08-28
> > **Response to authors**
> >
> > I appreciate the authors' effort put into tackling the raised issues and performing extra experiments. I think the current state of the paper is much better. For this reason, I have raised my score to 6.
> >
> > There are still certain aspects that can be improved, especially around i) ease of integration with other frameworks (modularity != integration), ii) scalability (twitter on simple LR model) and iii) privacy (lack of uniform integration with DP/SecAgg beyond Ditto). Last, I would urge the authors to revisit some of the initial reviews comments for improving the clarity and legibility of the text.
> > Nevertheless, I believe that the paper will overall benefit the community.
> >
> > Certain questions on the new experiments:
> > * Wrt Table 21, why is memory consumption that much different between the baselines? Are the authors measuring total memory consumption over all the workers cumulatively?
> > * How was the hyperparameter of the number of clusters selected in HypCluster?
> > * Wrt heterogeneity and fairness, there is also the possibility that certain models may not fit at all on certain clients [x,xx,y,yy] or that clients are selected in a biased manner [z,zz] based on their capabilities. It would be interesting to have a commentary/discussion on how these issues could affect fairness and personalised accuracy.
> >
> > [x] Horvath, S., Laskaridis, S., Almeida, M., Leontiadis, I., Venieris, S., & Lane, N. (2021). Fjord: Fair and accurate federated learning under heterogeneous targets with ordered dropout. Advances in Neural Information Processing Systems, 34, 12876-12889.
> > [xx] Diao, E., Ding, J., & Tarokh, V. (2020). HeteroFL: Computation and communication efficient federated learning for heterogeneous clients. arXiv preprint arXiv:2010.01264.
> > [y] Dudziak, L., Laskaridis, S., & Fernandez-Marques, J. (2022). FedorAS: Federated Architecture Search under system heterogeneity. arXiv preprint arXiv:2206.11239.
> > [yy] He, C., Annavaram, M., & Avestimehr, S. (2020). Group knowledge transfer: Federated learning of large cnns at the edge. Advances in Neural Information Processing Systems, 33, 14068-14080.
> > [z] Lai, F., Zhu, X., Madhyastha, H. V., & Chowdhury, M. (2021). Oort: Efficient federated learning via guided participant selection. In 15th {USENIX} Symposium on Operating Systems Design and Implementation ({OSDI} 21) (pp. 19-35).
> > [zz] Li, C., Zeng, X., Zhang, M., & Cao, Z. PyramidFL: A Fine-grained Client Selection Framework for Efficient Federated Learning.

---

> > > ### Author Response · Authors · 2022-08-29
> > > **Many thanks for your great suggestions!**
> > >
> > > We genuinely appreciate the reviewer for the constructive feedback! As the discussion period is reaching its end, we don't have enough time to make very detailed modifications and fully reflect on your suggestions in the paper. We will continue to improve our work in light of your comments as much as possible. Regarding the further questions, we provide some brief and quick answers here:
> > >
> > > 1. This difference in values is reasonable as we used the *standalone* simulation mode. Specifically, we used *only a single process for each set of experiments (one dataset and one method)* and monitored the memory consumption of the corresponding process. The monitoring was done by wandb, which tracked system metrics every 2 seconds, and we then counted the mean and peak values of these results for system metrics.
> > > Throughout the single set experiment, we actively free the Video RAM but keep the model objects resident in the main memory for fast running, which results in larger memory consumption for datasets involving a larger number of clients, and methods that maintain multiple models for each client.
> > >
> > > 2. We chose the number of clusters to be 3 for HypCluster to align with another baseline, FedEM, which also leverages multiple internal models to capture the relationships among clients. HypCluster explicitly allocates each client into one of the three clusters and uses one of the three models accordingly. By contrast, FedEM learns soft weights for each internal model and uses the ensemble of all three models in inference. When hard limits are introduced such that the weights of all clients are binary and still sum to 1, FedEM will become very similar to HypCluster.
> > >
> > > 3. Thanks for your insightful suggestion and the mentioned useful literature! In our experiment, we have simulated the federated training process with heterogeneous device resources in terms of both computation and communication.
> > > The newly considered resource heterogeneity does bring different observations, such as exacerbating the degree of unfairness in FL participation between clients, and slower global convergence.
> > > As for the other types of resource heterogeneity such as memory, we agree that it's very interesting to investigate how they will affect the fairness and personalized accuracy, and how to design pFL methods to handle corresponding new potential challenges. There are several promising directions, such as biased client selection [z,zz] and allowing different clients to use models of different sizes and even different architectures via techniques such as knowledge distillation [x, yy], sparsification [xx], Neural Architecture Search (NAS) [y], quantization [a] and matrix factorization [b].
> > > We will include the discussion in the final version and leave more in-depth explorations for future work.
> > >
> > > Thanks again for your dedicated effort to enhance our paper!
> > >
> > > --------
> > > Reference (besides the "x" ~ "zz" mentioned in the reviewer's reply)
> > >
> > > [a] FedMask: Joint Computation and Communication-Efficient Personalized Federated Learning via Heterogeneous Masking
> > >
> > > [b] Factorized-FL: Agnostic Personalized Federated Learning with Kernel Factorization & Similarity Matching

---

> ### Author Response · Authors · 2022-08-26
> **Responses to Reviewer Ytx8, "Correctness" to "Clarity" Part, page 4/5**
>
> > "I would count datasets with different LDA parameters uniquely"
>
> Thank you for your rigorous review! After adding the *Twitter* dataset (see above response to "Datasets are simply ... and small" in reply page 1/5), we currently have included 10 distinct datasets and 12 dataset variants. Following your suggestion, we have amended the revision accordingly: "xx datasets -> xx dataset *variants*".
>
> > Common baselines/setups may be trickier than offering a benchmarking suite
>
> Thanks for your detailed comments! We do agree that "Given that new work tends to emphasise on the positive results, the trend of presenting favourable results/setups where a new approach shines may not be avoided." But this trend can be exacerbated if there is a lack of a comprehensive benchmark, it is difficult for the community to evaluate a new approach from different metrics, scenarios and datasets at the same time. Noting no free lunch theorem, we would like to start by building a benchmark suite that is as comprehensive, easy to use and extensible as possible to make further research easier and more objective.
>
> > Global-Train method performs bad on FEMNIST
>
> Many thanks for your comments! We apologize for the error in the result of the Global-Train method in Table 1, which we have fixed from 52.48 to 74.51 in the revision. After re-running the method with the tracked config file, and double-checking the logs of historical experiment results, we found that the results of the experiment from one of the three runs at that time were wrong and somewhat out of date (before we fixed the bug Global-Train mode in this [pull request](https://github.com/yxdyc/FederatedScope/commit/7bc3eeb4a29d40ca8a4eb111c266ae077fe25759)). For now, we also double-checked the codes for the relevant experiment results processing and the logs for other experimental runs. Again, thank you very much for pointing this out!
>
>
> > Bottom accuracy being zero in table 4 means that it is essentially worse than random.
>
> Yes, you are right. This is because with heterogeneous partitioning, some clients have very small amounts of data (or even just a few samples), which results in that their models have not learned useful information and happened to make mistakes on all of the few test samples.
>
> > Clarity improvements.
>
> We sincerely thank you for the suggestions to further improve our presentation! We have made according modifications in the revision based on your suggestions, and list answers for your mentioned questions below:
> - "Are FLOPs counted as the amount of floating operations in training or inference?" The FLOPS are counted as the sum of amounts for both training and inference via a per-operator flops counting tool, [fvcore/flop_count](https://github.com/facebookresearch/fvcore/blob/main/docs/flop_count.md).
> - "what granularity are the communication bytes reported?" Yes, the reported communication bytes are counted over upstream+downstream across participants until convergence with early stopping.
> - "how the communication rounds were selected per technique here?" We employ early stopping with a large number of total FL rounds for all the experiments, as we mentioned in the last paragraph of Sec. 4.4.
>
>
> > About the Dockerfiles
>
> Thanks for the careful checking and valuable feedback! We have fixed the typo in the Dockerfile link (enviroment -> environment), and now the correct link is accessible (https://github.com/alibaba/FederatedScope/blob/master/environment/docker_files/federatedscope-torch1.8-application.Dockerfile). Besides, for the evaluation part, after installing the FederatedScope, we provide all reproducible config files via single-line running such as "python federatedscope/main.py --cfg benchmark/pfl_bench/yaml_best_runs/FedBN_FEMNIST-s02.yaml". For multinode configurations, we can simply and repeatedly start containers from the same built Docker image in different computation nodes, and then leverage the wandb sweep tool to conduct the HPO just as we introduced in the README. Thanks again!

---

> ### Author Response · Authors · 2022-08-26
> **Responses to Reviewer Ytx8, "Weakness" Part to "Correctness" Part, page 3/5**
>
>
> > The suite is based on FederatedScope, which might not be easily "intergratable" with other FL frameworks
>
> Many thanks for your comments! To demonstrate the extendibility, we newly integrate FedScale [1] into our benchmark via a simulator that executes the behaviors of clients according to virtual timestamps of clients' messages delivered to the server, such that we can evaluate performance with heterogeneous system dynamics and heterogeneous device information used by FedScale. More details are given in the above response to "Lack of support/integration of system heterogeneous FL". We would like to clarify that although the specific APIs of our codebase and other FL frameworks differ, our codebase and many other FL frameworks are both designed to be as modular as possible, and moreover, share some similarities in the flexible, event-driven programming philosophy to describing federation behaviors, such as FedScale and FedML. To further reduce the barriers to use and learning costs of our codebase, we also provide extensive documentation including [APIs](https://federatedscope.io/refs/index), [executable scripts](https://github.com/alibaba/FederatedScope/tree/master/scripts), customised [configurations](https://github.com/alibaba/FederatedScope/tree/master/federatedscope/core/configs) and [tutorials](https://federatedscope.io/docs/quick-start/).
>
>
> > Reproducibility w.r.t. client availability
>
> Thank you for your question! As a benchmark for pFL, we do value the reproducibility of experiments and put a lot of effort into ensuring this, including containerized environments, extensive experimental scripts with tracked random seeds, and detailed experiment logs. Specifically, we carefully fix the random seeds for the underlying packages that may introduce randomness (see this [function](https://github.com/alibaba/FederatedScope/blob/e3f541dba78891a140d8ffe5e984855c5a390775/federatedscope/core/auxiliaries/utils.py#L31)). To simulate the clients' availability, we employ random client sampling during the message broadcast stage, i.e., some clients may lose connection just as they are not sampled (see the [samplers](https://github.com/alibaba/FederatedScope/blob/e3f541dba78891a140d8ffe5e984855c5a390775/federatedscope/core/sampler.py)). Moreover, as the above response to "Lack of support/integration of system heterogeneous FL" shows, we employ a simulator with virtual timestamps, which are estimated by the deterministic cost model and hetero-device information from FedScale. These implementations allow our benchmark to support good reproducibility, even for cross-device scenarios with heterogeneous system resources.
>
>
> > Missing system metrics in runtime such as peak memory usage
>
> Thank you very much for your suggestion! Our benchmark indeed supports monitoring more runtime metrics. Thanks to the good integration with wandb, we have already monitored the usage of system resources in runtime including utilization of CPU, GPU, memory, disk, etc.
>
> In the below table and revision pdf (Appendix Table 21), we newly report the average and peak process memory usage (in MB) and process running times (in seconds). In general, most pFL algorithms do have higher time and space overheads.
> We omit to report results for other metrics since we started very many sets of experiments concurrently, taking up as much of the graphics card's memory and maximizing CPU/GPU utilization as possible, these metrics do not differ much from one of our different experiments. However, it is worth noting that they can be used to analyze algorithm efficiency, and system performance bottlenecks, and to optimize running speed in single-experiment scenarios.
>
> ||FEMNIST|||SST-2|||
> |---|---|---|---|---|---|---|
> ||MEM_avg|MEM_peak|T_run|MEM_avg|MEM_peak|T_run|
> |Global-Train|87|87|11|87|93|892|
> |Isolated|747|1352|1108|118|151|994|
> |FedAvg|10507|17707|840|297|400|319|
> |FedAvg-FT|13400|21752|1193|314|372|545|
> |FedProx|19389|20729|1415|6635|7378|672|
> |FedProx-FT|21209|22504|1935|7805|8161|1810|
> |pFedMe|1881|1980|7205|263|366|1572|
> |pFedMe-FT|18574|21332|4676|282|368|1080|
> |HypCluster|21660|22387|2803|6942|7510|678|
> |HypCluster-FT|22569|23589|3602|7624|8233|887|
> |FedBN|11135|15407|1011|280|373|332|
> |FedBN-FT|17347|25650|936|267|332|551|
> |FedBN-FedOPT|21867|35335|2881|216|311|202|
> |FedBN-FedOPT-FT|9827|14711|845|261|350|283|
> |Ditto|1128|1268|4628|149|205|638|
> |Ditto-FT|1111|1533|8915|229|292|599|
> |Ditto-FedBN|1116|1227|4049|197|229|593|
> |Ditto-FedBN-FT|1454|1730|8528|216|290|627|
> |Ditto-FedBN-FedOpt|1177|1274|5782|203|238|560|
> |Ditto-FedBN-FedOpt-FT|1299|1565|7545|275|334|637|
> |FedEM|940|1063|5070|267|358|1568|
> |FedEM-FT|564|619|7592|327|374|3210|
> |FedEM-FedBN|572|912|12051|270|337|1615|
> |FedEM-FedBN-FT|490|617|9391|307|368|3040|
> |FedEM-FedBN-FedOPT|756|842|8977|251|343|1314|
> |FedEM-FedBN-FedOPT-FT|607|639|19137|301|353|3116|

---

> ### Author Response · Authors · 2022-08-26
> **Responses to Reviewer Ytx8, "Weakness" Part, page 2/5**
>
>
> > Lack of support/integration of system heterogeneous FL.
>
> Many thanks for your comments! We are glad that the proposed pFL-Bench has good extensibility to support experiments in heterogeneous device resource scenarios, where clients have different computational and communication capacities.
> - Within the response period, we newly **integrate FedScale [1] into our benchmark** with a simulator that executes the behaviors of clients according to virtual timestamps of their message delivery to the server.
> - The virtual timestamps are updated by the estimated execution time based on **clients' computational and communication capacities** with the cost model proposed in FedScale.
> - The server employs an over-selection mechanism for clients at each broadcast round, and thus some clients' messages may be dropped since the clients have different system capacities and different response speeds corresponding to [real-world mobile devices](https://github.com/SymbioticLab/FedScale/tree/master/benchmark/dataset/data/device\_info).
>
> Here we take the Ditto on FEMNIST as an example and present the results with heterogeneous device resources below and in our revision.
> - Let $s$ be the clients sampling rate for each FL round, and $s_{agg}$ be the minimal ratio of received feedback w.r.t. the number of clients for the server to trigger federated aggregation in over-selection mode.
> For the homo-device case, we set $s=0.2$ and for the hetero-device case, we set the $s=0.25$ and $s_{agg}=0.8$, leading to the same number of clients used for each aggregation.
> - From the results, we find that the hetero-device version has slower convergence speeds ($T'=0$ indicates that early-stopping is not triggered within the large number of FL rounds $T=1000$), and it gains worse performance than the homo-device version, especially for the bottom accuracy ($\breve{Acc}$) and standard deviation of the average accuracy ($\sigma$).
> This shows unfairness among clients due to the fact that some low-resourced clients have too long computation or communication time to make their feedback incorporated into the federated aggregation, calling for more considerations w.r.t. device heterogeneity within pFL algorithm design.
>
> |  | $\overline{Acc}$ | $\widetilde{Acc}$ | $\Delta$ | $\overline{Acc}'$ | $\sigma$ | $\breve{Acc}$ | FLOPS | Com. |  $T'$    |
> |--|---|-|--------|-|---|-|--|-------|------|
> | Ditto, Homo-Device   | 88.39  | 2.2             | -86.19 | 87.18           | 7.52   | 78.23           | 849.3G | 2.81M | 610  |
> | Ditto, Hetero-Device | 79.76          | 1.43            | -78.33 | 77.39           | 11.25  | 61.76           | 1.72T  | 5.72M | 0    |
>
>
> > Lack of support/integration of techniques from privacy realms.
>
> Many thanks for your comments! We agree that "The tradeoff dynamics between personalisation, privacy and fairness are the most interesting and underexplored in the field." Currently, we *have made many privacy-related fundamental components and algorithms available in FederatedScope*, such as [Differential Privacy (DP)](https://federatedscope.io/docs/dp/) and various [privacy attacks method](https://federatedscope.io/docs/privacy-attacks/) to examine the privacy-preserving strength.
> Further cross-research among pFL and privacy will be convenient based on the extensible modular design of our benchmark.
>
> As a preliminary example, here **we demonstrate the combination of the pFL with a Differential Privacy algorithm**, the nbafl [2] that achieves $(\epsilon, \delta)$-DP via noise injection and gradient clipping.
> We list the test accuracy below and plot the learning curves of FedAvg and Ditto on FEMNIST dataset with various ($\epsilon, \delta$)-DP in the revision (Appendix, Figure 9).
>
> We can see that generally, with larger $\epsilon$ and $\delta$, the accuracy ($\overline{Acc}$) is better for both the compared methods, which meets our expectations for privacy protection that the stronger the protection, the more performance degradation there is.
> Interestingly, Ditto shows significantly better robustness for the dramatically varying privacy protection strengths during the whole learning process than FedAvg.
> This may be because, in the noise perturbation scenarios, the personalized local model potentially brings up more local optimal points that can be reached for clients.
> But there is still a gap for the best achievable performance between Ditto and FedAvg, leaving an interesting open question about how to reduce the degree of performance degradation with co-design of personalization and noise injection.
>
> | $\epsilon$ | $\delta$ | FedAvg | Ditto |
> |--|--------|--------|-------|
> | 10       | 0.01   | 0.08   | 0.67  |
> | 10       | 0.17   | 0.32   | 0.67  |
> | 10       | 0.76   | 0.57   | 0.67  |
> | 50       | 0.01   | 0.7    | 0.67  |
> | 50       | 0.17   | 0.76   | 0.68  |
> | 50       | 0.76   | 0.8    | 0.72  |
> | 100      | 0.01   | 0.79   | 0.7   |
> | 100      | 0.17   | 0.81   | 0.73  |
> | 100      | 0.76   | 0.82   | 0.73  |

---

> ### Author Response · Authors · 2022-08-26
> **Responses to Reviewer Ytx8, "Weakness" Part, page 1/5**
>
>
> We sincerely thank you for your appreciation and helpful review and make the following responses:
>
> > felt at times more like a survey rather than a benchmark
>
> Thanks for the comments!
> - We would like to clarify that to make the paper easier to read and to make it as easy and comprehensive as possible for readers from different backgrounds (e.g. those unfamiliar with FL or pFL) to understand pFL, we have set up two brief Sections, Sec. 2 "Related Work" and 3 "Background and Problem Formulation, which may cover information somewhat similar to what would be covered in a survey.
> - However, as a benchmark paper, we have devoted a fair amount of space to introduce baselines, datasets, evaluation protocols, extensions and designs of proposed benchmark, as well as extensive experimental results and analysis.
> As a more specific example, compared with the pFL survey paper ["towards personalized federated learning"], we significantly differ from it with Sec. 4 "Benchmark Design and Resources" and Sec. 5 "Numerical Results and Analysis".
>
> > Datasets are simply gathered from pre-existing literature and small
>
> Thank you for your comments!
> - As a benchmark for pFL, we do not over-claim that our contribution is introducing *new* datasets, but including numerous and public FL datasets that cover a wide range of domains, scales and partition manners for comprehensive evaluation, and providing easy extension for other datasets and further research. We provide modular implementation and unified interfaces for easy data processing, and the proposed code-base is compatible with a large number of datasets from other popular DataZoos such as LEAF, Torchvision, Huggingface datasets and FederatedScope-GNN. As an example, we can easily add the *Federated Netflix* dataset that includes *480,189* clients with natural and heterogeneous user splitting as shown in this [pull-request](https://github.com/alibaba/FederatedScope/pull/281).
> - Furthermore, we newly conduct experiments on the **Twitter** dataset from LEAF for sentiment analysis task. We adopt a subset which contains **13,203** users, the partition manner for this dataset is natural w.r.t. users rather than by LDA, and the median number of data samples per client is 7. We include a more detailed introduction and visualization in the revision pdf. Here we list experimental results below and report the full and more readable results in the revision. Interestingly, on this highly Non-IID dataset, the *Isolated* method and fine-tuning technique show good performance since users have very small local data size and diverse textual expression patterns. Besides, the average accuracy results weighted by local data size ($\overline{Acc}$) are larger than the uniform averaged ones ($\overline{Acc}'$), and the bottom accuracy ($\breve{Acc}$) is 0 in most cases, indicating that the clients having small numbers of local data still have large room to improve their performance.
>
> |  | $\overline{Acc}$ | $\widetilde{Acc}$ | $\Delta$ | $\overline{Acc}'$ | $\sigma$ | $\breve{Acc}$ | FLOPs | Com. | $T'$ |
> |---|---|---|---|---|---|---|---|---|---|
> | Global-Train | 55.56 | - | - | 55.56 | - | - | 31.19G | - | 20.33 |
> | Isolated | 70.04 | - | - | 67.44 | 37.86 | 0 | 36.3M | - | 0 |
> | FedAvg | 62.15 | 61.24 | -0.91 | 56.98 | 37.97 | 0 | 726.45K | 60.32K | 223.33 |
> | FedAvg-FT | 70.53 | 71.17 | 0.64 | 68.45 | 36.19 | 0 | 4.75M | 19.26K | 70.67 |
> | FedOpt | 62.09 | 61.64 | -0.45 | 59.95 | 37.88 | 0 | 726.45K | 60.32K | 220.33 |
> | FedOpt-FT | 71.08 | 71.41 | 0.33 | 69.2 | 36.28 | 0 | 4.52M | 18.35K | 66.67 |
> | pFedMe | 63.45 | 62.52 | -0.94 | 58.18 | 36.67 | 0 | 603.46K | 29.3K | 106.67 |
> | pFedMe-FT | 84 | 71.8 | -12.2 | 78.82 | 33.28 | 22.22 | 12.07M | 15.57K | 53.33 |
> | Ditto | 70.23 | 49.6 | -20.63 | 66.9 | 38.05 | 0 | 2.64M | 38.42K | 140.33 |
> | Ditto-FT | 69.99 | 51.32 | -18.67 | 66.91 | 38.08 | 0 | 4.2M | 36.59K | 133.33 |
> | FedEM | 63.44 | 62.68 | -0.75 | 61.7 | 37.72 | 0 | 18.82M | 95.64K | 163.33 |
> | FedEM-FT | 70.97 | 71.59 | 0.62 | 70.19 | 36.35 | 0 | 24.19M | 35.27K | 40.57 |

---

### Official Review · Reviewer_mZSd · 2022-07-28
**Impressive workload, insights unclear, clarification questions**

**Rating:** 5
**Confidence:** 4
**Correctness:** Good
**Clarity:** Good.

**Strengths:**

I am impressed by the general workload of performing experiments on multiple datasets and baseline methods, while also having some concerns on the settings and insights from the benchmark.

**Weaknesses:**

The decisions on some of the datasets seem to be ad-hoc: why sample 5% of EMNIST clients, why use \alpha={5,0.5,0.1} for CIFAR-10 and split it to 100 clients?
A related issue, I would strongly recommend the authors follow [Wang 2021 A Field Guide to Federated Optimization section 3.1] to discuss the practical setting, e.g., cross-device vs. cross-silo. The number of clients in the datasets in table 1 are relatively small; the authors mentioned client sampling, but I am not sure if it is always enforced, or only for EMNIST.

The insights of combining various methods are not quite clear. I appreciate the extensive experiments, but table 2 and table 3 are hard to interpret: Some methods are not personalization specific, such as FedOPT, if it consistently improves the performance, maybe it should be used by default. It is also not clear which version of FedOPT is used, is it FedADAM? Some methods like Ditto and FedBN assumes each client maintain their local states, it is not clear to me how they can be applied to unseen clients.


Clarification questions:
For \bar{Acc}, is there a training/validation partition on each client? What is the percentage of data for training?
The repository https://github.com/alibaba/FederatedScope/tree/master/benchmark/pFL-Bench seems to only provide scripts for EMNIST.

Minor:
I am not sure how useful Eq (1) is. The presentation looks problematic to me.
It seems to be a variant of the population risk where the objective of the global model is decomposed to ERM for training G and validation R. The relationship between population risk and ERM is not entirely new, for example, see Eq (1) & (2) in  [Wang 2021 A Field Guide to Federated Optimization]. It is confusing that personalization only happens for C, not unseen clients \bar{C}.


**Additional Feedback:**

NA.

**Documentation:**

Yes.

**Ethics:**

No concerns.

**Relation To Prior Work:**

Yes.

**Summary And Contributions:**

This is a benchmark paper studies personalization in the federated learning (FL) setting. In FL, users/clients jointly train a model without sharing their personal local data. In personalized FL (pFL), each client can learn their own model in addition to the global model. This paper provides empirical study on ~10 datasets (sampled EMNIST, 3 CIFAR-10, COLA, SST-2, Cora, Pubmed, Citeseer, MovieLens1M and MovieLens10M), and (including the combination) multiple baseline methods (FedAvg, FineTune, pFedMe, FedBN, Ditto, FedEM).


======================== After discussion
I still appreciate the workload of this draft. However, I also have concerns about the choices of datasets, and presentation/discussion of FL algorithms, see https://openreview.net/forum?id=2ptbv_JjYKA&noteId=ICd1jefTRoW for details.

---

> ### Author Response · Authors · 2022-08-26
> **Responses to Reviewer mZSd, Part 1/2**
>
> Many thanks for your appreciation and helpful review! We make the following responses:
>
> > Datasets settings
>
> Thank you for your questions!
> - We would like to clarify that the main motivation of choosing these settings for the datasets is to cover as wide range of domains, scales and partition manners and as easy-to-use for a comprehensive evaluation as possible. The adopted number of clients ranges from a few (cross-silo case such as graph datasets) to thousands (cross-device case such as recommendation datasets), and the choice of $\alpha$ is from some previous pFL works [1,2]. For the mentioned client sampling, we enforced it for all the experimental datasets except the graph datataset containing a few clients.
> - As to why some of these datasets are subsets of the original version, this is mainly due to considerations of efficiency and ease of reproducibility. It is worth noting that simulation with pFL algorithms on a large client scale is very challenging, due to the fact that we need to **maintain distinct (personalized) model object for each client**. Let's take the FEMNIST as an example, which has 3,550 users and suppose we adopt the widely-used two-layer CNN network. Although this model only occupies ~200MB, maintaining 3,550 such models would consume more than ~700GB memory.
> Different from non-personalized FL algorithms, for which it is feasible to maintain only one model object for all the clients, for pFL algorithm, we may have to switch and cache the personalized models among CPU, GPU and even disks. Due to the large number of baselines and datasets included in our benchmark, and the corresponding huge hyper-parameter search space, we used several subsets of the FL datasets to reduce the reproduction and experimental barriers.
>
> > Relatively small datasize
>
> Our benchmark supports larger datasets thanks to the good extendibility with FederatedScope. To further enrich adopted datasets, we newly conduct experiments on the **Twitter** dataset from LEAF for sentiment analysis. We adopt a subset containing **13,203** users, the partition manner for this dataset is natural w.r.t. users rather than by LDA, and the median number of data samples per client is 7. We include more detailed introduction and visualization in the revision pdf. Here we list experimental results below and report the full and more readable results in the revision. Interestingly, on this highly Non-IID dataset, the *Isolated* method and fine-tuning technique show good performance since users have very small local data size and diverse textual expression patterns. Besides, the average accuracy results weighted by local data size ($\overline{Acc}$) are larger than the uniform averaged ones ($\overline{Acc}'$), and the bottom accuracy ($\breve{Acc}$) is 0 in most cases, indicating that the clients having small numbers of local data still have large room to improve their performance.
>
> |  | $\overline{Acc}$ | $\widetilde{Acc}$ | $\Delta$ | $\overline{Acc}'$ | $\sigma$ | $\breve{Acc}$ | FLOPs | Com. | $T'$ |
> |---|---|---|---|---|---|---|---|---|---|
> | Global-Train | 55.56 | - | - | 55.56 | - | - | 31.19G | - | 20.33 |
> | Isolated | 70.04 | - | - | 67.44 | 37.86 | 0 | 36.3M | - | 0 |
> | FedAvg | 62.15 | 61.24 | -0.91 | 56.98 | 37.97 | 0 | 726.45K | 60.32K | 223.33 |
> | FedAvg-FT | 70.53 | 71.17 | 0.64 | 68.45 | 36.19 | 0 | 4.75M | 19.26K | 70.67 |
> | FedOpt | 62.09 | 61.64 | -0.45 | 59.95 | 37.88 | 0 | 726.45K | 60.32K | 220.33 |
> | FedOpt-FT | 71.08 | 71.41 | 0.33 | 69.2 | 36.28 | 0 | 4.52M | 18.35K | 66.67 |
> | pFedMe | 63.45 | 62.52 | -0.94 | 58.18 | 36.67 | 0 | 603.46K | 29.3K | 106.67 |
> | pFedMe-FT | 84 | 71.8 | -12.2 | 78.82 | 33.28 | 22.22 | 12.07M | 15.57K | 53.33 |
> | Ditto | 70.23 | 49.6 | -20.63 | 66.9 | 38.05 | 0 | 2.64M | 38.42K | 140.33 |
> | Ditto-FT | 69.99 | 51.32 | -18.67 | 66.91 | 38.08 | 0 | 4.2M | 36.59K | 133.33 |
> | FedEM | 63.44 | 62.68 | -0.75 | 61.7 | 37.72 | 0 | 18.82M | 95.64K | 163.33 |
> | FedEM-FT | 70.97 | 71.59 | 0.62 | 70.19 | 36.35 | 0 | 24.19M | 35.27K | 40.57 |

---

> > ### Comment · Reviewer_mZSd · 2022-08-29
> > **Thanks for the new experiments, not convinced for the dataset choice**
> >
> > I appreciate the new experiments, but I am not convinced on the dataset choice.
> >
> > I appreciate the number of datasets as I mentioned in my original comment, however, it is possible to create arbitrary number of new federated learning datasets by artificially partition a centralized dataset to decentralized. It is not clear to me what insights we can get from such datasets and experimental results.
> >
> > I appreciate the honesty and understand the resources constraints. However, I think having to maintain a local model in memory is a design choice of algorithms (or even just the framework implementation), not a requirement for FL or pFL. The number of clients in cross-device FL can easily scale up to millions (Advances and Open Problems in Federated Learning, table 1), and stateless algorithms are preferred (A Field Guide to Federated Optimization, section 3.1). I am not asking for an academic paper to run experiments on millions of clients (though it has been done in several previous papers, for example, FedScale: Benchmarking Model and System Performance of Federated Learning at Scale), but proper discussion of possible practical application setting should be included. Even for stateful algorithms, the resource constraint sounds like an implementation issue of the framework, I do not understand why all models have to be maintained in memory; maybe a database on hard disk can be considered?
> >
> > I have concerns that the choice of dataset in this benchmark is overfitting to the current state of algorithms instead of providing playground for future algorithmic design. There are concurrent preprints that clearly discussed application settings (FLamby: Datasets and Benchmarks for Cross-Silo Federated Learning in Realistic Settings ), scales to larger number of clients with stateless algorithms (Motley: Benchmarking Heterogeneity and Personalization in Federated Learning). I do not expect the authors to know these work before the submission, but hope to provide concrete suggestions by examples.

---

> > ### Comment · Reviewer_mZSd · 2022-08-29
> > **Client sampling**
> >
> > Could the authors provide direct answer to how clients sampling is done? Apologize if I missed it in the draft.

---

> ### Author Response · Authors · 2022-08-26
> **Responses to Reviewer mZSd, Part 2/2**
>
> > About the FedOPT and un-seen clients for Ditto and FedBN
>
> Thanks for your questions!
> - We implement FedOPT that supports various server optimizers (via specifying the config *cfg.fedopt.optimizer.type* in our code). We used the SGD as the server optimizer in our experiments similar to [3,4,5], and found that it did not consistently improve the performance, especially for the models containing batch-normalization or layer-normalization. As the original FedOPT paper did not discuss the special aggregation handling of BN/LN related parameters, we did not add additional modifications for this case in order to respect the original approach. We have empirically found that aggregation of BN/LN parameters on the server side can lead to unsuitable local BN/LN parameters, and severe performance degradation in several cases.
> - Out of similar respect for the original Ditto and FedBN papers, which did not discuss how to apply to unseen clients, we have kept the behaviors of their algorithms in inference. That is, after receiving the global model maintained by the server based on seen clients, we fine-tune the unseen clients' local models (if there is a combination with the FT technique) according to their original training behaviors, and finally use their local models for inference.
>
>
> > Clarification questions for bar{Acc}, training percentage and scripts
>
> Thank you for your questions! For all the adopted datasets, the train/val/test splitting is conducted within the local data of each client. And we list the specific train/val/test ratio of all datasets in our appendix. For the provided scripts, we clarify that in the Section 3 of the README "Run the experiments", we mentioned that the since the searching scripts and best config yaml files for all experiments involve about 600 files and 6w+ code lines, we omit them and present an example for FEMNIST here. The full scripts can be downloaded from a packed small zip file maintained in Alibaba Cloud, in which we organize the scripts for all the methods and all the datasets as multiple directories named by dataset name. Thanks again!
>
> > Minor issues on Equation (1)
>
> Many thanks for your comments and suggestion! We modified the $R$ term into $R\big(\{\mathbb{E}\_{(x, y) \sim D\_j} [f(\theta\_j; x, y) |\theta\_g] \}\_{j\in\tilde{C}} \big)$ in our revision, which indicates that the personalization with local model $\theta\_j$ also happens for the unseen clients $\mathcal{\tilde{C}}$ but conditioned on the global model trained on seen clients $\theta\_g$.
>
>
> **References**
>
> [1] Personalized Federated Learning using Hypernetworks. ICML 2021.
>
> [2] Bayesian Nonparametric Federated Learning of Neural Networks. ICML 2019.
>
> [3] Measuring the effects of non-identical data distribution for federated visual classification. Arxiv 2019.
>
> [4] SlowMo: Improving Communication-Efficient Distributed SGD with Slow Momentum. ICLR 2020.
>
> [5] FedNLP: Benchmarking Federated Learning Methods for Natural Language Processing Tasks. NAACL 2022 Findings.

---

### Official Review · Reviewer_3xnj · 2022-07-28
**A comprehensive benchmark suite for Personalized FL**

**Rating:** 7
**Confidence:** 3
**Clarity:** The paper is well-written.

**Strengths:**

1. Numerous relevant datasets and use cases
2. Extensive evaluation
3. Good documentation

**Weaknesses:**

- The datasets are relatively small in size (i.e. the number of clients.) and do not have a natural split.

**Additional Feedback:**

Please refer to the section above.

**Correctness:**

The experiments are sound and the evaluation methods are designed appropriately and performed correctly.

**Documentation:**

The dataset is hosted via Github and seems to be well-maintained. The documentation is well-written.

**Ethics:**

No ethical concerns were identified.

**Relation To Prior Work:**

The paper discusses the difference from previous work and is fairly comprehensive.

**Summary And Contributions:**

pFL-bench is a comprehensive benchmark suite for personalized FL (pFL). It contains more than 10 datasets in diverse application domains and more than 20 pFL baseline implementations. The paper also demonstrates the utility of pFL-bench by conducting an extensive experiment.

---

### Author Response · Authors · 2022-08-26
**Summary of the revision**

Dear Reviewers,

We sincerely thank you for your time and efforts in reviewing our paper as well as the constructive and insightful feedback! We carefully read and responded to your comments. During the response period, we have conducted additional experiments and modifications in our newly uploaded revision, which can be summarized as follows:

- We have added experiments on a larger-sized NLP dataset, Twitter from LEAF, which contains 13,203 clients and the partition manner for this dataset is natural w.r.t. users.
- We have implemented one more non-pFL method (FedProx), one more clustering-based pFL method (HypCluster), and conducted more experiments for them.
- We have supported the simulation of heterogeneous system dynamics via integrating FedScale with its provided heterogeneous device information and have conducted demonstrative experiments.
- We have supported the exploration of trade-offs between pFL and privacy protection techniques and conducted demonstrative experiments with Differential Privacy.
- We have added results about more system metrics in runtime and more probing for data Non-IID degree (the clients' pairwise similarity of their feature or label distributions in terms of Jensen–Shannon distance).
- We have improved the paper's presentation to make it more clear and more accurate according to the reviewers' questions and suggestions.

Thanks again for your detailed and helpful suggestions! We believe the paper has been further greatly improved with the additional experimental results and modifications.

---

### Meta-Review · Area_Chair_uoRQ · 2022-09-07

**Recommendation:** Accept
**Confidence:** 5

**Metareview:**

This work highly benefited from the rebuttal period, and, following the reviewers comment, I now recommend it to be accepted. It offers a nice overview of personalized FL. After checking, I would however concur with the reviewers in asking the authors to be particularly careful with the given code that could benefit from an extra documentation and a maintenance in time  — adding more flexibility with other FL frameworks.

---

### Decision · Program_Chairs · 2022-09-16

Accept